# Elevated atmospheric CO$_2$ decreases methylmercury production in freshwater lakes

Pei Lei [1,2,3,15], Jin Zhang[1,2,15], Ri-Qing Yu[4], Maciej Bartosiewicz[5,6], Chengjun Li [7], R. Iestyn Woolway [8], Martin Tsz-Ki Tsui [9], Tao Jiang[10], Bo Meng[11], Raymond W. M. Kwong [12], Yuming Guo [13], Huan He[1], Xinghui Xia [14], Hongqiang Ren[2] & Huan Zhong [2] ✉

Elevated atmospheric carbon dioxide (CO$_2$) level reshapes microbial communities in nature, yet its consequences for neurotoxic methylmercury (MeHg) production in waters remain unclear. Here, we show that elevated CO$_2$ levels (650 and 1000 ppm) consistently reduced net MeHg production across 45 freshwater lakes spanning 1200 longitudinal kilometers, particularly in eutrophic conditions (54–96%). Elevated CO$_2$-driven shifts in carbon substrates favored hydrogenotrophic methanogens (e.g., *Methanobacterium*) lacking the *hgcA* methylation gene over *hgcA*-harboring acetoclastic strains (e.g., *Methanosarcina*), decreasing methanogen abundance (18–98% in *hgcA* copies) and activity (13–53% in net CH$_4$ production) and suppressing Hg methylation. Model simulations predict a 33%–74% global decline in lake MeHg production under future CO$_2$ scenarios, partially counteract MeHg increases associated with intensified algal blooms under warming. This overlooked pathway highlights the need to integrate interacting climate drivers to improve predictions of MeHg risks in a climate-changing future.

The biogeochemical cycling of the global pollutant mercury (Hg) is closely linked with methanogenesis[1–3], a primary process for organic matter decomposition that produces the potent greenhouse gas (GHG) of methane (CH$_4$)[4]. This connection is mainly due to methanogens, which are pivotal in methanogenesis and meanwhile act as essential microbes for Hg methylation[5], converting inorganic mercury

(IHg) into neurotoxic methylmercury (MeHg). Consequently, changes in methanogens and methanogenesis significantly impact the MeHg production in natural waters and its subsequent accumulation in aquatic food chains, where MeHg can biomagnify up to ten million times[6] and lead to a demonstrable decrease in newborn intelligence quotient worldwide[7]. Particularly in eutrophic lakes, the

[1]School of Environment, Nanjing Normal University, Nanjing, China. [2]State Key Laboratory of Water Pollution Control and Green Resource Recycling, School of Environment, Nanjing University, Nanjing, China. [3]State Key Laboratory of Lake and Watershed Science for Water Security, Nanjing Institute of Geography and Limnology, Chinese Academy of Sciences, Nanjing, China. [4]Department of Biology, Center for Environment, Biodiversity and Conservation, The University of Texas at Tyler, Tyler, TX, USA. [5]Department of Environmental Sciences, University of Basel, Basel, Switzerland. [6]Institute of Geophysics, Polish Academy of Sciences, Warsaw, Poland. [7]Institute of Environmental Research at Greater Bay Area, Guangzhou University, Guangzhou, China. [8]School of Ocean Sciences, Bangor University, Bangor, UK. [9]School of Life Sciences, The Chinese University of Hong Kong, Hong Kong SAR, China. [10]Interdisciplinary Research Centre for Agriculture Green Development in Yangtze River Basin, College of Resources and Environment, Southwest University, Chongqing, China. [11]State Key Laboratory of Environmental Geochemistry, Institute of Geochemistry, Chinese Academy of Sciences, Guiyang, China. [12]Department of Biology, York University, Toronto, ON, Canada. [13]School of Public Health and Preventive Medicine, Monash University, Melbourne, VIC, Australia. [14]State Key Laboratory for Water Environment Simulation, School of Environment, Beijing Normal University, Beijing, China. [15]These authors contributed equally: Pei Lei, Jin Zhang. ✉e-mail: zhonghuan@nju.edu.cn

decomposition of algal organic matter (AOM) increased the availability of labile carbon and $CH_4$ production by 16–44%, which was accompanied by a sharp spike in MeHg levels (1–2 orders of magnitude), primarily driven by methanogens[8]. Understanding the shifts in methanogen-mediated MeHg production in lakes is thus essential for predicting future risks of this potent neurotoxin in aquatic systems and its adverse impacts on wildlife and human health.

The composition and functions of methanogenic communities are impacted by the ambient levels of carbon dioxide ($CO_2$), a primary GHG with increasing atmospheric concentrations[9,10]. Crucially, this influence potentially extends to microbial MeHg production in water, which is a major source of MeHg in human food chains[11]. However, the cascading effects of elevated $CO_2$ on methanogenic communities in natural waters and hence on microbial MeHg production remain unclear. Evidence shows that elevated $CO_2$ levels increase the abundance of methanogens, with a 31–156% rise in rice paddies and 63–98% in marshes, leading to a 28–120% increase in $CH_4$ emissions[12,13]. Nevertheless, these changes may not directly apply to other natural waters, including freshwater lakes, due to differences in methanogen communities and their metabolic types across aquatic systems[4], with distinct capacities for Hg methylation[14]. For example, the acetoclastic methanogens, which use acetate as an energy substrate, predominate in agricultural wetlands[15], whereas hydrogenotrophic methanogens use $H_2/CO_2$, are more prevalent in freshwater lakes[16], yet their Hg methylation capacities differ[17]. Understanding how microbial MeHg production responds to elevated $CO_2$ is further complicated by the presence of a wide array of microbial groups, apart from methanogens, that co-contribute to MeHg formation, including sulfate-reducing bacteria (SRB) and iron-reducing bacteria (FeRB)[18]. In consequence of this complexity, existing knowledge of changes in methanogens under elevated $CO_2$ levels may not reliably predict concomitant changes in microbial MeHg production and its bioaccumulation in human food chains amid climate change.

Here, we aim to elucidate the effects of elevated $CO_2$ on Hg methylation in freshwaters, which are essential to aquaculture production, currently accounting for 77% of edible aquaculture production globally[19]. Particularly in Asia, human MeHg exposure was dominated by freshwater aquatic food intake, in addition to seafood[20]. Meanwhile, freshwater lakes are well-recognized hotspots for microbial MeHg production, a process significantly enhanced by eutrophication, which generates AOM upon algal decomposition[21] and thus provides carbon sources that boost the growth of Hg-methylating methanogens[8]. We thus quantified the net changes of MeHg in response to elevated atmospheric $CO_2$ levels (650 ppm and 1000 ppm) across 45 freshwater lakes spanning 1200 longitudinal kilometers along the Yangtze River in China. Results indicate that rising $CO_2$ levels consistently lead to a decline in both the abundance of Hg-methylating methanogens and net MeHg production. Global simulations, facilitated by our newly compiled dataset, reveal a crucial pathway through which the interplay of climatic factors helps reduce uncertainties in predicting MeHg risks in natural waters amid climate change. These discoveries deepen our understanding of global changes in distribution and risks of this potent neurotoxin under future scenarios.

## Results and discussion

### Reduced MeHg production in freshwater lakes under elevated $CO_2$

We demonstrate that elevated $CO_2$ concentrations inhibit MeHg production in freshwater lakes. This is achieved by quantifying net MeHg production in 45 freshwater lakes under various $CO_2$ and AOM levels (see Supplementary Table 1 for lake characteristics). As algal proliferation increases in response to nutrient enrichment, their biomass commonly serves as an indicator of nutrient abundance and eutrophication[22,23]. Therefore, microcosm experiments involving the addition of algal biomass to simulate eutrophic conditions provide a

practical approach to studying MeHg production in eutrophic waters. Under ambient $CO_2$ levels, AOM markedly enhanced MeHg production, yielding 1–3 orders of magnitude higher levels across 45 lakes (Supplementary Fig. 1), likely driven by stimulated methanogenesis-mediated MeHg production[8,24]. Under the projected $CO_2$ level of 1000 ppm in the year 2100, corresponding to the high GHG emissions scenario (SSP5-8.5) outlined in the Sixth Assessment Report from Intergovernmental Panel on Climate Change[25], however, we observed a substantial and prevalent decrease in net MeHg production across the surveyed freshwater lakes. The reduction ranged from 14 to 96%, with an average decrease of 63% compared to ambient 420 ppm $CO_2$ conditions ($p < 0.05$, Fig. 1 and Supplementary Fig. 1). Similar trends were also evident for other projected $CO_2$ levels in 2100 (i.e., 650 ppm, corresponding to the SSP2-4.5 scenario with intermediate GHG emissions), where net MeHg production showed reductions ranging from 45 to 61% (Fig. 2a and Supplementary Fig. 2a).

Particularly, decline of MeHg production under elevated $CO_2$ levels was most pronounced under eutrophic conditions, simulated as high algal biomass additions. The most obvious decrease in net MeHg production (by 54–96%) was observed in microcosms incubated at 1000 ppm $CO_2$ and with added algal biomass (Fig. 1b). Furthermore, the $CO_2$-induced effect remained consistent across different amounts of AOM (i.e., "+200 Algae" to "+1000 Algae"), resulting in a reduction of MeHg from 26 to 93% ($p < 0.05$, Fig. 2b and Supplementary Fig. 2b). During the long-term (60-day) of algal decomposition in lake water with added algal biomass, the inhibition ranged from 60 to 93% with averaging 80% (Fig. 2c and Supplementary Fig. 2c).

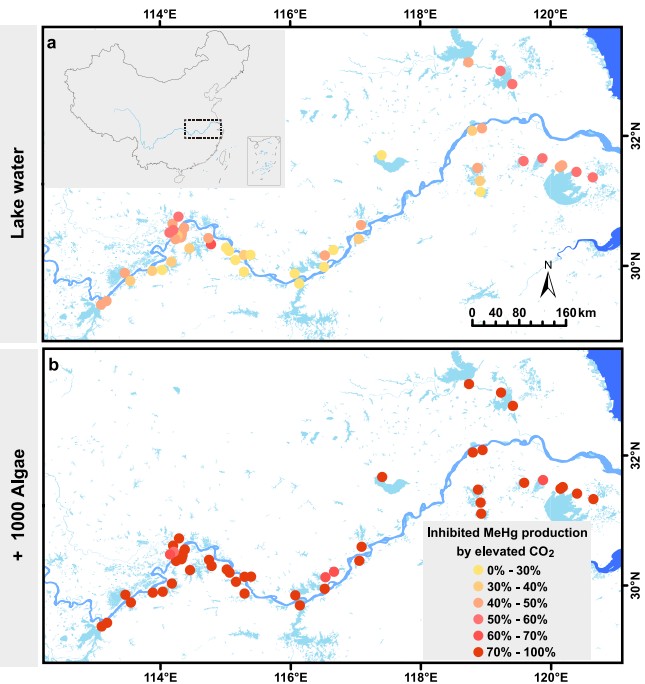

**Fig. 1 | Elevated $CO_2$-inhibited MeHg production in lakes along the middle and lower reaches of the Yangtze River in China. a** "Lake water": unfiltered lake water. **b** "+1000 Algae": unfiltered lake water amended with algal biomass to simulate bloom (resulting in approximately 1000 μg $L^{-1}$ chlorophyll $a$) in Experiment A. Inhibited MeHg production by elevated $CO_2$ is denoted as the percentage changes of MeHg levels between "Ambient $CO_2$" and "Elevated $CO_2$." The map of the middle and lower reaches of the Yangtze River was generated using 1:1,000,000 National Fundamental Geographic Information Data of China (https://www.webmap.cn/commres.do?method=result100W). The inset map in the upper-left corner was produced on the basis of a licensed basemap (Map Approval Number: GS(2016) 1569) provided by the Ministry of Natural Resources of China (http://bzdt.ch.mnr.gov.cn/).

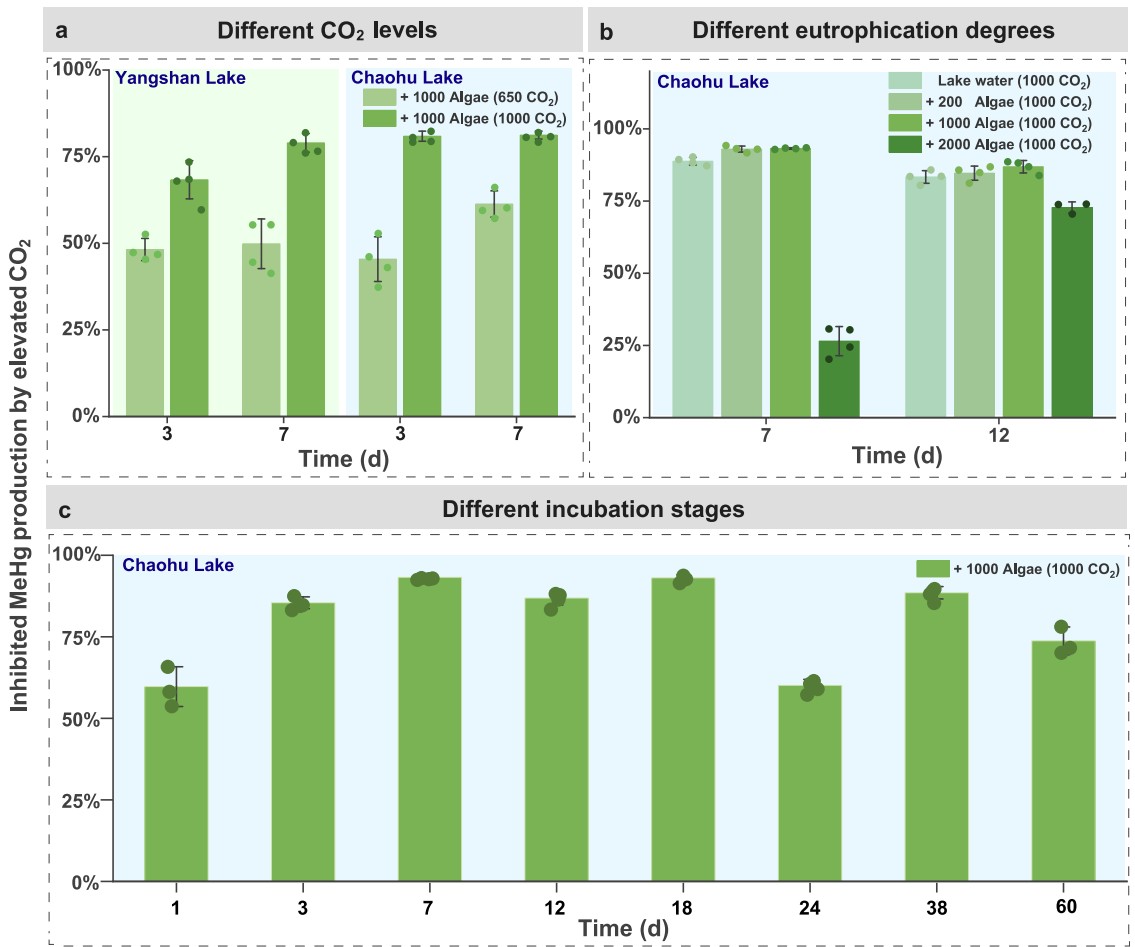

**Fig. 2 | Elevated CO₂-inhibited MeHg production under different scenarios.**
**a** Inhibition rate of MeHg production under different elevated CO₂ levels (650 or 1000 ppm) compared to ambient 420 ppm CO₂. **b** Inhibition rate of MeHg production by elevated CO₂ (1000 ppm) compared to ambient CO₂ (420 ppm) under bloom intensities (200–2000 µg L⁻¹ chlorophyll *a*). **c** Inhibition rate of MeHg production by elevated CO₂ (1000 ppm) compared to ambient CO₂ (420 ppm) under different stages of algal decomposition. "Lake water": unfiltered lake water. "+200 Algae," "+1000 Algae," and "+2000 Algae": unfiltered lake water (40 mL) amended with 0.001, 0.005, and 0.01 g of algal biomass, respectively. Before sampling on "Day 0," lake water was equilibrated for 4 h with Hg(II). Inhibited rate of MeHg production by elevated CO₂ are denoted as the percentage changes of MeHg levels between "Ambient CO₂" and "Elevated CO₂." Error bars show standard errors of the mean (*n* = 3 or 4). Each replicate is depicted as a circle on the bar, and details of each replicate can be found in the Source data.

We further extended the environmental relevance of elevated CO₂-impacted MeHg dynamics (see Supplementary Text 1). The CO₂-driven suppression of microbial Hg methylation is independent of background Hg levels, as shown by spiking experiments across a broad inorganic Hg (IHg) gradient (2–500 ng L⁻¹). Elevated CO₂ consistently reduced net MeHg production relative to ambient CO₂ levels, with 21–33% inhibition in unamended lake water and 66–83% in algal biomass-added treatments (Supplementary Fig. 3). We also established DOM gradients reflecting different autochthonous: allochthonous ratios (100:0 to 0:100; AOM vs. Soil organic matter, SOM). Results confirmed that AOM was a stronger driver of methylation than SOM, consistent with previous studies[8,26], and that elevated CO₂ significantly inhibited net MeHg production only when AOM comprised ≥50% of the DOM pool (*p* < 0.001; Supplementary Fig. 4), underscoring the need to clarify how DOM composition modulates CO₂-driven MeHg inhibition. Moreover, under combined climate forcing (elevated CO₂ and/or warming, Δ*T* = 4.4 °C, SSP5-8.5 by 2100), MeHg production inhibition under "Elevated CO₂" and "Warming × Elevated CO₂" treatments (26–76% reduction) did not differ significantly (*p* > 0.05; Supplementary Fig. 5), indicating that the inhibitory effect of elevated CO₂ outweighs the stimulatory impact of warming, thus supporting the robustness of our model in projecting global MeHg trends. Importantly, such inhibition was also

observed in other freshwater systems (e.g., 79% decrease in pond water; 75% in river water, Supplementary Fig. 6), suggesting that elevated CO₂ may mitigate MeHg risks across global inland waters. In summary, elevated CO₂ exerts a robust suppression of microbial MeHg production that is independent of background Hg levels, dependent on DOM sources, overrides warming effects, and extends broadly across freshwater systems.

Our results expand understanding of how elevated CO₂ reshapes Hg biogeochemistry and the associated risks of MeHg production, going beyond previous considerations of the impacts of CO₂ elevation on Hg toxicity[27], Hg bioaccumulation[28], and global Hg budgets[29]. To date, there has been relatively little focus on understanding the effect of elevated CO₂ on microbial MeHg production[30,31], the primary source of MeHg in human food chains. This knowledge gap is particularly critical in freshwater lakes, known as hotspots for MeHg formation[8,21] and sensitive to climate changes[32]. While recent studies indicate that AOM derived from algal decomposition in eutrophic lakes boosts microbial MeHg formation and inhibits its photodegradation[8,33], our study shows that such increases of MeHg levels in eutrophic waters would potentially be counteracted by future elevation in atmospheric CO₂ levels and thus inhibited Hg methylation. Implications of these processes are further discussed in the later sections.

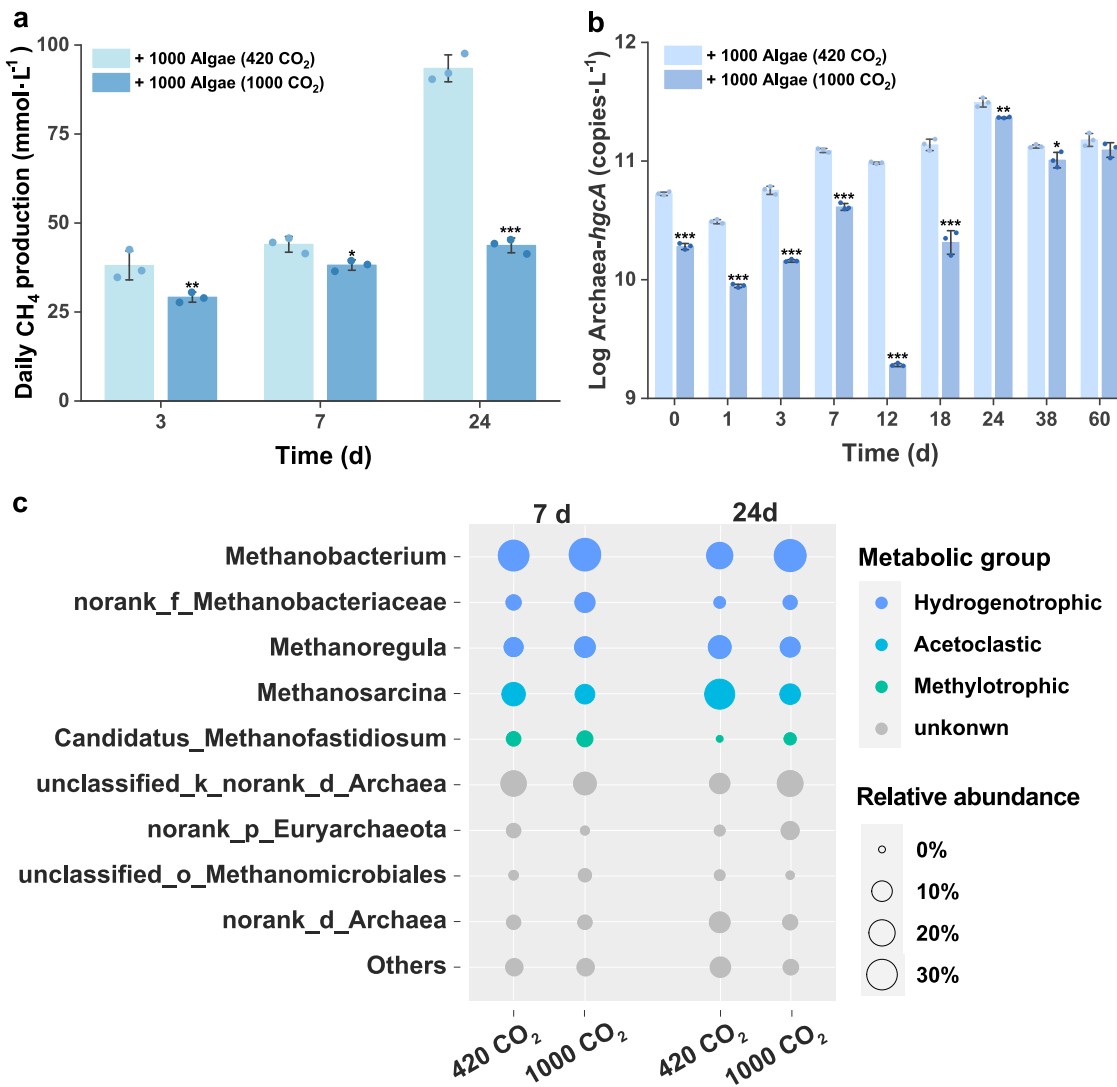

**Fig. 3 | Activity, abundance, and metabolic types of methanogens are impacted by elevated $CO_2$. a** Daily production of $CH_4$, **b** abundance of Archaea-*hgcA* genes, and **c** relative abundance of methanogens during the periods of algal decomposition under ambient $CO_2$ (420 ppm) or elevated $CO_2$ (1000 ppm) levels. Error bars show standard errors of the mean ($n = 3$).

## The underlying mechanism by which elevated $CO_2$ levels inhibit Hg methylation in eutrophic waters

Given the more pronounced decreases in net MeHg production in eutrophic waters under elevated $CO_2$, we then focus on elucidating the mechanisms of such decreases under eutrophic conditions. Theoretically, the decrease in net MeHg production could be caused by decreased Hg methylation and/or increased MeHg demethylation[18]. Using Hg stable isotope tracing technique, we found that Hg methylation rates ($k_m$) were 83–89% lower under elevated $CO_2$ than under ambient conditions in both natural or eutrophic waters ($p < 0.001$), whereas MeHg demethylation rates ($k_d$) did not differ significantly between treatments (Supplementary Fig. 7). These complementary lines of evidence demonstrate that the observed decrease in net MeHg production lake water under elevated $CO_2$ was primarily attributed to inhibited Hg methylation, rather than the increases in MeHg demethylation. Consequently, we focused on how the dominant methylating microbes respond to elevated $CO_2$.

In eutrophic lakes, methanogens have been recognized as the key microbes for Hg methylation due to their higher relative abundances and increased activities facilitated by low-molecular weight AOM derived from algal decomposition[8]. Their enhanced activity significantly contributes to Hg methylation[34]. Therefore, changes in

methanogens under elevated $CO_2$ offer a compelling explanation for the corresponding shifts in MeHg production in eutrophic waters. Our investigations demonstrate a substantial impact of elevated $CO_2$ on methanogenic activity, evidenced by the net daily $CH_4$ production rates. In eutrophic waters, these rates were considerably suppressed, averaging a 30% decrease with a range between 13 and 53% in comparison to the control under ambient $CO_2$ (Fig. 3a). Furthermore, the qPCR assays demonstrate that elevated $CO_2$ results in a decrease in copies of *hgcA*−as the Hg methylation gene−of Archaea by 18–98% (Fig. 3b), which are strongly correlated with the $CO_2$-inhibited MeHg production ($R^2_{adjust} = 0.866$, $p < 0.001$, Supplementary Fig. 8). These various pieces of evidence imply that methanogenetic activity and abundance are lowered under elevated $CO_2$, which provides an explanation for the inhibited microbial Hg methylation in eutrophic waters, where methanogens play a dominant role in MeHg production[8].

We further demonstrate that the observed MeHg decrease in eutrophic waters under elevated $CO_2$ could be explained by the shifting abundance of different metabolic types of methanogens, with an increase in weak or non-Hg methylating types and a decrease in strong Hg methylating counterparts. *Methanobacterium* spp., the most dominant group of methanogens in this study, is generally known as

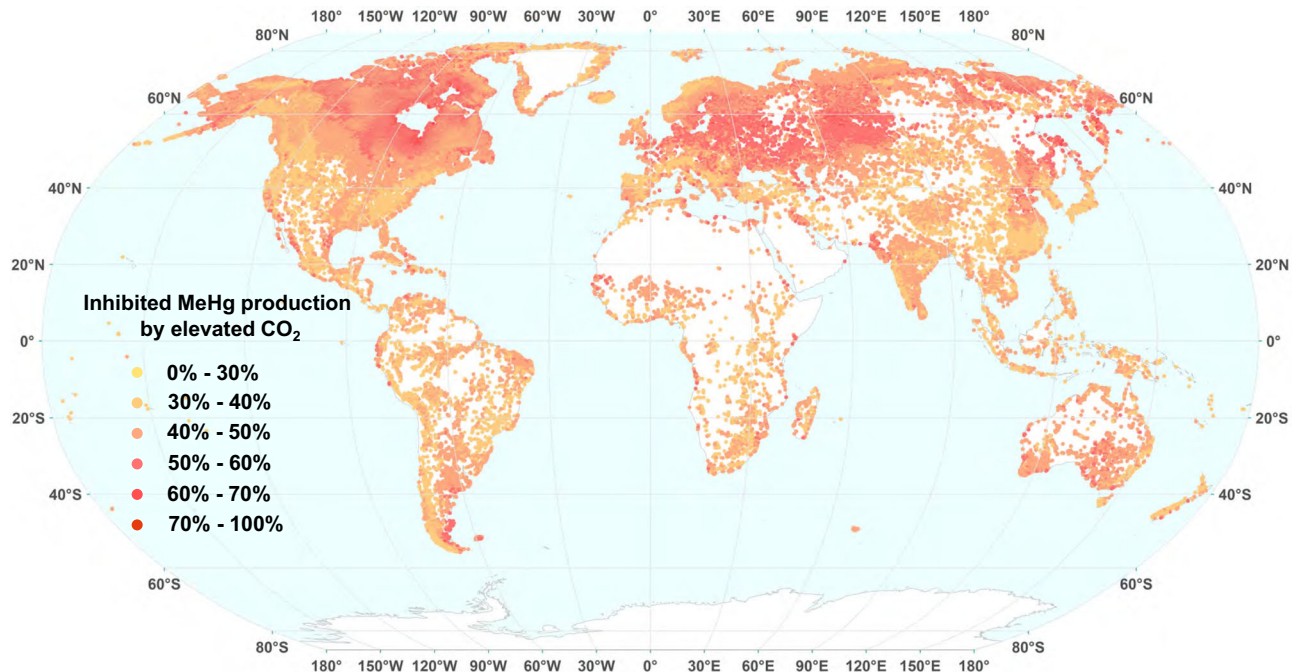

**Fig. 4 | Projected decreases in MeHg production under elevated $CO_2$ conditions (1000 ppm in 2100) in global lakes.** The predicted DOC concentrations in global lakes with a surface area larger than 0.1 km² were used (1,427,496 lakes in total, ranging from 0.0002 to 27.0 mg L⁻¹, with an average of 9.24 mg L⁻¹), and the database was obtained from a recent report[51]. The relationship between DOC and MeHg was adapted from a recent global meta-analysis[52]. The global map was obtained from Natural Earth dataset (https://www.naturalearthdata.com/downloads/10m-physical-vectors/).

hydrogenotrophic[35,36], while *Methanosarcina* is primarily known as acetoclastic methanogen[37]. High-throughput sequencing and bioinformatic analysis of *mcrA* genes revealed that elevated $CO_2$ leads to a notable shift in the methanogenic community: a decrease in the relative abundance of aceticlastic methanogens (e.g., *Methanosarcina* reduced from 29% in ambient $CO_2$ to 10% under elevated $CO_2$ on Day 24) and an increase in hydrogenotrophic methanogens (e.g., *Methanobacterium* increased from 22% in ambient $CO_2$ to 35% under elevated $CO_2$ on Day 24, Fig. 3c). Additional BLAST searches of *HgcAB* orthologs, indicates that among the 51 *HgcAB* orthologs identified in *Euryarchaeota*, 88% of potential Hg-methylating methanogens are associated with *Methanomicrobia*[38]. Within this taxon, primarily as the acetoclastic methanogen, *Methanosarcina* was observed in our samples. In contrast, hydrogenotrophic *Methanobacterium* spp., are much less likely to carry *hgcA* genes necessary for Hg methylation[38,39]. Consequently, we propose that the concomitant increase in hydrogenotrophic non-Hg methylating methanogens and the decrease in aceticlastic Hg methylating methanogens contributed to the reduced net MeHg production under elevated $CO_2$. This hypothesis requires further confirmation through a more quantitative comparisons of Hg methylation abilities across different methanogenetic taxa. Additionally, the interaction between methanogens and syntrophs, which has not been examined in this study, might play a role in modulating Hg methylation activities[40].

Such a transition from aceticlastic to hydrogenotrophic methanogens is ascribed to the changes in the predominant carbon sources under elevated $CO_2$ levels. In eutrophic lakes, the decomposition of planktonic biomass produces low-molecular-weight organic matter, e.g., aromatic proteins, stimulating aceticlastic methanogens[8]. These methanogens are possibly highly effective at utilizing bioavailable carbon and thrive across a broad range of environmental niches, including eutrophic lakes[41,42], which makes them key contributors in microbial MeHg production under eutrophic conditions[8]. However, elevated $CO_2$ levels, known as the preferred carbon source of hydrogenotrophic methanogens[43], might facilitate the growth of this metabolic type of methanogens, leading to a compositional shift from acetoclastic methanogens to hydrogenotrophic methanogens in eutrophic waters, a shift likely attributable to $CO_2$-stimulated fermentative $H_2$ production that relieves $H_2$ limitation and promotes hydrogenotrophic dominance (see detailed discussion in Supplementary Text 2). Similarly, elevated $CO_2$ increased the relative abundance of hydrogenotrophic methanogens (e.g., *Methanobacterium* and *Methanocella*) while significantly reducing the relative abundance of acetoclastic methanogens (e.g., *Methanosaeta*) in rice paddies[12]. An increase in the relative abundance of hydrogenotrophic methanogens was also reported in temperate marsh microcosms when exposed to elevated $CO_2$ levels[43].

In particular, the inhibited MeHg production under elevated $CO_2$ is unlikely due to other major Hg methylating microbes, such as SRB or FeRB, or possible changes in Hg bioavailability under elevated $CO_2$ due to acidification. Further discussion is presented in Supplementary Text 3.

Methanogens are widely distributed in natural environments where MeHg production occurs, including wetland soils, peatlands, freshwater, or marine sediments[44–46]. The key role of methanogens in Hg methylation, as well as their sensitivity to climatic factors, including elevated $CO_2$, global warming, and water acidification[10,47,48], highlight the urgency to elucidate and consider methanogen-mediated MeHg production in future climate change scenarios. Our results provide the initial evidence that the effects of climate change on methanogens would in turn affect microbial MeHg production, particularly in eutrophic waters subject to algal blooms, which are increasing under global warming[49,50]. Global implications of these findings are estimated and discussed in the following section.

### Estimating how elevated $CO_2$ reduces MeHg levels in global lakes

To estimate the potential impact of the projected $CO_2$ elevation in 2100 (i.e., 1000 ppm[25]) on MeHg levels in global lakes, we coupled a relationship between elevated $CO_2$ and inhibited net MeHg production

in lakes with different ambient DOC contents, as established in this study (see Methods for details), and existing global datasets on (1) predicated[51,52] or (2) observational DOC and MeHg levels (summarized in Supplementary Data 1) in global lakes. Our simulations, based on global distribution of predicted DOC and MeHg, suggest that elevated $CO_2$ could lead to decreases in MeHg levels by 33–74% (average 47%) in lakes worldwide (Fig. 4). More specifically, in lakes characterized by high ambient DOC levels (averagely 18.7 mg L$^{-1}$) and high MeHg contents (averagely 0.61 ng L$^{-1}$), accounting for approximately 2.5% of the lakes examined, MeHg levels are projected to drop by 60–74% as $CO_2$ levels rise. Approximately one-third of the global lakes are anticipated to experience a reduction in MeHg levels between 30 and 45%, another third between 45 and 50%, whilst the remaining third will witness a decrease exceeding 50%. Similarly, inputting observed ambient DOC and MeHg concentrations–based on literature data (Supplementary Data 1)–into the same model predicts that elevated $CO_2$ could lead to a decrease in MeHg concentrations in lakes across the globe, ranging from 34 to 79%, with an average of 47%. Uncertainty analysis by Monte Carlo simulation shows that elevated $CO_2$ inhibits MeHg production in lakes worldwide, with an uncertainty range of 36–68% (Supplementary Text 4).

While the global reduction in MeHg levels under elevated $CO_2$ is estimated based on our modeling and found to be evident in lakes, it is important to note that there are numerous other factors that may influence the Hg methylation process in real-world scenarios, adding uncertainty to our estimates. More field studies quantifying the responses of MeHg production to changes in multiple climatic factors, including but not limited to $CO_2$ levels, are needed to further improve the reliability, and reduce the uncertainty of predicting MeHg levels in lakes in the context of climate change.

## Environmental impacts: the interplay of climatic factors helps reduce uncertainties in MeHg risk prediction

Excessive nutrient inputs to lakes have led to a globally accelerated eutrophication, which in turn affects the magnitude, frequency, and duration of algal blooms[53]. Recent laboratory tests and field observations indicate that AOM in eutrophic lake waters promotes MeHg formation while reducing its photodegradation[8,33]. This contributes to MeHg accumulation in eutrophic lakes and increases the associated risks to wildlife and humans. Nevertheless, our findings demonstrate a counterbalancing effect from future climate change conditions: elevated atmospheric $CO_2$ mitigates methanogen-mediated Hg methylation in eutrophic waters, potentially offsetting the risks of global warming-amplified algal blooms[54] and hence enhanced Hg methylation[8].

Currently, predictions concerning Hg biogeochemistry and associated risks in the context of climate change commonly concentrate on the influence of individual climatic factors, such as rising $CO_2$[55] or global warming[56]. However, the interactive effects of multiple drivers are rarely assessed, partly because of the limited understanding of how the Hg cycle reacts to multiple simultaneous controlling processes. Our research highlights the potential for biased predictions concerning future Hg risks when considering a single climatic factor. This is because various climatic factors can exert conflicting influences on Hg biogeochemical cycles, as exemplified by the counteracting effects of $CO_2$ elevation and global warming–and hence algal blooms–on microbial MeHg production. Such a previously overlooked interplay of climatic factors can play a crucial role in stabilizing MeHg levels in natural settings amidst climate change. These combined effects should be incorporated into regional or global models of Hg biogeochemical cycles. Failing to acknowledge this unrecognized self-stabilization mechanism could lead to an overestimation of the impacts of climate change on MeHg risks (further detailed in Supplementary Text 5).

Interestingly, our experimental results indicate that the decrease in MeHg production under elevated $CO_2$ levels may extend from lakes to other freshwater systems, e.g., 79% decrease for pond water and 75% decrease for river water (see Supplementary Fig. 6). This implies that elevated $CO_2$ in the future could potentially mitigate MeHg formation on a broader scale, possibly affecting global inland freshwater systems that supply food to local populations and provide other critical ecosystem services[57]. Consequently, it is essential to address the crucial knowledge gaps related to the distinct and combined impacts of diverse climate facets on the production, accumulation, and transfer of MeHg in aquatic ecosystems. These insights serve as the foundation for developing more nuanced models of the Hg biogeochemical cycle, aiding in the prediction of the regional and global risk of MeHg to wildlife and humans.

More importantly, incorporating the combined effects of multiple climatic factors on the Hg cycle is key to reducing uncertainty in evaluating the effectiveness of our ongoing global Hg mitigation efforts. Currently, evaluating the effectiveness of Hg emission reductions under the Minamata Convention on Mercury is hindered by a significant disparity: while Hg emissions have been reduced, human exposure to MeHg has decreased at a much slower rate[7]. Per unit reduction in Hg emissions, human exposure has been estimated to decline by only one-third to one-half, reflecting the influence of complex processes such as legacy Hg in environmental reservoirs and bioaccumulation lags in food chains. This observed "mismatch"– where exposure decreases less than expected from emission reductions–likely arises from such multi-faceted drivers, including remobilization of legacy Hg by natural and anthropogenic factors[58].

Our study identifies a distinct climatic pathway that further complicates this emission–exposure relationship: elevated $CO_2$ suppresses microbial MeHg production in lakes, leading to reduced aquatic MeHg levels regardless of emission changes. This pathway would amplify the decline in human exposure beyond direct emission reductions, and should be accounted for when evaluating the effectiveness of global Hg mitigation efforts and interpreting temporal trends in biotic MeHg concentrations. Thus, our findings do not resolve the current mismatch, but rather reveal a previously unrecognized climate–microbe–MeHg interaction that may shape future exposure risks. Specifically, future increases in atmospheric $CO_2$ may counteract warming-driven enhancements of algal blooms, thereby reducing uncertainties in predicting aquatic MeHg risks under climate change. Incorporating this climate–microbe–Hg nexus into regional and global models of the biogeochemical cycling of Hg is crucial for reducing the uncertainty in the prediction of environmental MeHg concentrations under climate change scenarios. Such an improvement will substantially improve our ability to assess the effectiveness of the Minamata Convention on Mercury in mitigating MeHg exposure under the complex influence of multiple climatic factors.

## Methods
### Field sample collection
To examine the impacts of elevated $CO_2$ on MeHg production in lakes, we selected 45 freshwater lakes (No.1–45; 48 sampling sites, Supplementary Fig. 9 and Supplementary Table 1) spanning 1200 longitudinal kilometers in China. We also conducted laboratory microcosm experiments to characterize the kinetics of MeHg levels and the shifts in Hg methylating microbes under both ambient and elevated $CO_2$ conditions. These water samples were collected exclusively from the lakes along the middle and lower reaches of the Yangtze River, which constitute 28.6% of China's population and contribute 20.9% to the national Gross Domestic Product in 2018[59]. Many of these lakes experience anthropogenic eutrophication[60]. Lake water samples were gathered during the period from August to September in 2022.

At each sampling site, approximately 500 mL surface water samples were collected in triplicate at 0.5 m below the lake surface. Samples for Hg and nutrient analyses were preserved with 0.1% HCl, and the fresh unfiltered water was used for all microcosm experiments. All

water samples were stored in a dark, cold room (4 °C) until processing/analysis. Algal biomass samples were collected during the algal bloom period in August of 2022 from Chaohu Lake (31°42′48.94″N, 117°21′48.37″E), the fifth largest freshwater lake in China[8]. Briefly, at the lake, algal biomass was collected using a plankton net and stored at 4 °C until processing. Before experiments, algal biomass was washed, freeze-dried, and mortar-homogenized, and then stored at −20 °C prior to use. Samples for total mercury (THg), MeHg, and major nutrients were analyzed following established protocols (see Supplementary Fig. 10 for results).

## Experimental microcosm setup

A series of microcosm experiments were designed to elucidate the effects and mechanisms of elevated $CO_2$ on microbial MeHg production in lake water. Specially, Experiments A to C aimed to reveal the general phenomenon of elevated $CO_2$ on microbial MeHg production under different conditions, such as different eutrophication scenarios or elevated $CO_2$ levels, by using diverse lake waters. Moreover, Experiment D was designed to elucidate the mechanisms of elevated $CO_2$-impacted MeHg production, and Experiment E was designed to explore whether elevated $CO_2$ affects the MeHg demethylation process. At the same time, MeHg levels were also determined for all processed incubation vials. The abundance of the hgcA genes in the mixture was quantified using the clade-specific degenerate primer pairs ORNL-Delta-HgcA and ORNL-Archaea-HgcA for Deltaproteobacterial methylators (e.g., SRB or FeRB) and Archaeal methylators (e.g., methanogens)[61], respectively (Supplementary Text 6). The quantification method for the relative abundances of acetoclastic to hydrogenotrophic methanogens was adopted from previous work (see Supplementary Text 7). Details of all treatments are summarized in Supplementary Table 2 and Text 8.

## Determination of MeHg levels in water

The levels of dissolved MeHg were determined following the GEO-TRACES protocol handbook[62]. The process involves adding 2 mL of the filtered sample to a 40 mL brown-glass bottle and adjusting the pH to 4 using a 2 M citrate buffer. A 2 mL sample volume was selected because, after the 7-day incubation, MeHg concentrations exceeded $0.2 \, \text{ng L}^{-1}$ in all treatments. Within the calibration curve range of 0.5–50 pg, this volume ensured accurate quantification. Subsequently, 50 μL of $NaBEt_4$ is added to the mixture. The solutions are then diluted with deionized water and measured using an automatic MeHg analyzer (CVAFS, Brooks Rand Model III, Brooks Rand Laboratories, USA) following Method 1630. Concentrations of MeHg were obtained using external calibration curves containing no less than 5 points ($r^2 > 0.999$). The detection limit for MeHg determinations was calculated as three times the standard deviation, the derived value being $0.01 \, \text{ng L}^{-1}$. During the analyses of MeHg samples, quality assurance and quality control (QA/QC) measures were implemented, including the use of blanks, blank spikes, sample duplicates, and matrix spikes. Levels of MeHg were measured following daily calibration with a MeHg stock solution (MeHgCl source, $1 \, \mu\text{g mL}^{-1}$ in 0.5% HOAc, 0.2% HCl, 30 mL, Brooks Rand Laboratories, USA). Blank samples of ultrapure water and lake water from CHL without or with algal biomass addition were spiked with MeHg at an initial concentration of $10 \, \text{ng L}^{-1}$ ($n = 5$ for all treatments). The recovery percentages of MeHg standard analyses were 93% ± 7.3% and 96% ± 9.8% for the blank (ultrapure water) and lake water samples, respectively. For lake water added with AOM, the MeHg standards showed a recovery of 99% ± 10.9%. The recoveries of all spikes (ranging from 79 to 122%) fell within the acceptable range as outlined in the EPA Method 1630 (i.e., between 65 and 135% for MeHg, Supplementary Fig. 11).

## Estimation of global MeHg reduction in lake water under elevated $CO_2$

To predict MeHg changes in response to elevated $CO_2$-induced MeHg production in water, we developed a global model for MeHg in lake water. This model relies on the observed drop in the MeHg production under elevated $CO_2$, as quantified in this study. Firstly, using data from Experiment A, we established that the relative reduction in MeHg production under elevated $CO_2$ in 45 lakes increases linearly with the rise of ambient DOC and MeHg contents (see Supplementary Table 3). Given that DOM in lake water has multiple sources (autochthonous and allochthonous), and that DOM from different sources exerts variable inhibitory effects on Hg methylation under elevated $CO_2$ (Supplementary Fig. 4), we incorporated this effect into our model predictions: for eutrophic waters, a mixture of autochthonous and allochthonous organic matter (AOM: SOM = 80%: 20%) was adopted in the model. Indeed, the decline in MeHg production rate (%) under elevated $CO_2$ can be calculated as: $R$ (Inhibited rate of MeHg (%) by elevated $CO_2$) = (0.015 ± 0.003) × [DOC] + (0.01 ± 0.001) × [MeHg] + (0.333 ± 0.027), $r^2_{\text{adjust}} = 0.701$, $p < 0.001$, [DOC] < 30 mg $L^{-1}$, where ±SE is the standard error of the fit and is considered one of the uncertainties in the model and $r^2_{\text{adjust}}$ is the correlation coefficient of the relationship. The above relationships are statistically significant ($p < 0.001$). The DOC in "Lake water + Algae" treatments can be estimated as: $[\text{DOC}]_{\text{Lake water + algal biomass}}$ = (1.389 ± 0.075) × $[\text{DOC}]_{\text{lake water}}$ + (6.4 ± 0.638) ($r^2_{\text{adjust}} = 0.991$, $p = 0.003$, see Supplementary Fig. 12a).

Subsequently, we estimated global MeHg reduction under elevated $CO_2$ using two independent databases: (1) global data on the predicated DOC[51] and MeHg levels[52] or observational ambient DOC and MeHg levels in global lakes. For the first one, predicted DOC concentrations in global lakes with a surface area larger than $0.1 \, \text{km}^2$ were used (1,427,496 lakes in total, ranging from 0.0002 to $27.0 \, \text{mg L}^{-1}$, with an average of $9.24 \, \text{mg L}^{-1}$)[51]. Using the relationship between DOC and MeHg from Lavoie et al.[52], we calculated the MeHg concentrations of 1,427,496 lakes globally: $[\text{MeHg}]_{\text{Asia}}$ = 0.004 × [DOC] −0.022; $[\text{MeHg}]_{\text{Europe}}$ = 0.015 × [DOC] + 0.156; $[\text{MeHg}]_{\text{North America}}$ = 0.045 × [DOC]−0.158. The global average slope and intercept were used to represent the relationship between MeHg and DOC concentrations in lakes located in other continents (including South America, Oceania, and Africa), i.e., $[\text{MeHg}]_{\text{others}}$ = 0.029 × [DOC] −0.019. The average concentration of calculated MeHg was $0.27 \, \text{ng L}^{-1}$. This dataset covers sites spanning six continents and a broad latitudinal range. Eventually, MeHg concentrations in global lake water bodies were predicted at the anticipated $CO_2$ concentration of 1000 ppm in 2100, utilizing the formula and data.

For the second database, we systematically reviewed literature spanning the period from 2000 to 2023 that documented MeHg and DOC levels in lakes (Web of Science™ search on June 6, 2023). From the search results of 201 scientific papers, we obtained 49 articles containing 229 samples for analyses (see Supplementary Data 1). These data were selected according to specific criteria: (i) inclusion of only field sampling data; (ii) acquisition of both unfiltered and filtered water samples; (iii) utilization of data from the most recent year collected at the same sampling point, and (iv) restriction of DOC concentrations to no more than $30 \, \text{mg L}^{-1}$. More specifically, this dataset comprises 74 water samples that exclusively display unfiltered total MeHg values, 88 water samples exclusively exhibiting filtered MeHg values, and 67 water samples providing both unfiltered and filtered MeHg values. Notably, linear regression analyses revealed a significant relationship between filtered and unfiltered total MeHg levels present in water: $[\text{MeHg}]_{\text{filtered}}$ = (0.708 ± 0.015) × $[\text{MeHg}]_{\text{unfiltered}}$−(0.001 ± 0.004), $r^2_{\text{adjust}} = 0.972$, $p < 0.001$ (see Supplementary Fig. 12b). The fitting errors mentioned above were considered as the uncertainty of the model. Accordingly, filtered MeHg concentrations in the 74 water samples lacking such data were calculated using the formula above. The average concentrations of DOC and MeHg in 229 lake water samples were $9.51 \, \text{mg L}^{-1}$ and $0.14 \, \text{ng L}^{-1}$, respectively. This dataset covering lakes spanning five continents was used to predict MeHg concentrations under the future $CO_2$ concentration (1000 ppm).

## Data availability
Source data are provided with this paper.

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

## Acknowledgements

This work was supported by the National Natural Science Foundation of China (NSFC; Grant Nos. 42525710 to H.Z., 42477399 and 42107383 to P.L.) and by the State Key Laboratory of Lake Province and Watershed Science for Water Security (Grant No. 2024SKL012 to P.L.). The authors gratefully acknowledge Shizhong Lei, Zhuoran Li, Dr. Sheng Guo, and Dr. Chao Wang for their valuable assistance with field water sample collection.

## Author contributions

P.L. and H.Z. conceptualized the study and provided overall supervision. J.Z. developed the methodology and prepared the figures. P.L., R.Y., M.B., and R.W. contributed to data validation. P.L. and J.Z. conducted the investigation. P.L., J.Z., and C.L. performed data curation. P.L. and J.Z. drafted the original manuscript. P.L. and H.Z. acquired the funding. M.T., T.J., B.M., R.K., Y.G., H.H., X.X., H.R., and H.Z. contributed to result interpretation and reviewed the manuscript.

## Competing interests

The authors declare no competing interests.
