## [Peer Review file · Nature Communications]

Elevated atmospheric CO₂ decreases methylmercury production in freshwater lakes

Corresponding Author: Professor Huan Zhong

Version 0:

Reviewer comments:

Reviewer #1

(Remarks to the Author)

Lei et al. present a detailed study on the relationship between atmospheric CO₂ and methylmercury production in freshwater lakes. It is a novel and interesting work that shows a possible pathway of reduced methylmercury under global warming, especially related to the decrease in *hgcA* genes. However, we have some concerns about its methods and arguments regarding their key findings on inhibited MeHg production and projected reduced MeHg levels in global lakes.

Major concerns:

The implications of the experimental design with no-light and constant temperature conditions need to be considered in the model extrapolation the results to lakes globally. "All vials were incubated in the dark at 25C to mimic the prevalent degradation conditions in lakes and preclude MeHg photodegradation" (supplementary information L326-328). To project MeHg dynamics in global lakes, the influence of light, but especially temperature changes need to be mentioned. A significant rise in CO₂ concentration will drive a marked increase in temperature. How do you exclude the potential significant temperature influence on methylation and demethylation activities in global lakes when estimating the reduction in MeHg levels caused by elevated CO₂? High temperature can likely stimulate microbial activities, including Hg methylation. The paper could still be published with the existing data and model, but the implications of not considering light and temperature need mentioned.

Furthermore, the global model on MeHg in lake water only uses DOC and MeHg as the parameters. The experiment, however, focuses on Algal Organic Matter (AOM), but DOC is used by the authors use to model the global trend of MeHg levels. The fact that AOM is only a fraction of DOC, and DOC is more complex regarding methylation and demethylation of Hg also needs to be acknowledged as a limitation of the model. .

Finally, the concluding lines of the paper state that its findings "offer a potential explanation for this mismatch". I think that statement is incorrect. The mismatch, as the authors present it, is that the reductions in emissions have only resulted in a decrease in human exposure that is proportionally smaller than emission reductions. The findings of the present paper suggest that even without any change in emissions, the MeHg in water would be going down as CO₂ levels rise, thus reducing human exposure even if Hg emissions were not reduced. This would tend to make human exposure decline faster than the Hg reductions. The authors should reconsider what they claim for this paper in regards to explaining the current mismatch between Hg emission reductions and human exposure.

Minor comments:

1. The abstract section on hydrogenotrophic and acetoclastic methanogens does not talk about differences in the *hgcA* genes between the two types of methanogens. It would be clearer if it did.
2. The North arrow needs to be indicated on the map Figure 1A.
3. Only data from Chaohu and Yangshan Lakes are presented in Figure 2. In supplementary Figure 2, Xuanwu Lake data is added to demonstrate the negligible impact of elevated CO₂ levels on demethylation. How about other lake data? Are they available in the supplementary information?
4. In Figure 2, only %MeHg data is shown. How about the MeHg concentrations?

Reviewer #2

(Remarks to the Author)

Reviewer #3

(Remarks to the Author)

I am very impressed by this extensive research study and the thorough effort to address potentially confounding factors in the supplemental material (demethylation, FeRB/SRB, etc). There was a lot of great research performed and a lot of novel findings. However, I believe that the context of the publication needs to be reconsidered – this is water column methylmercury production and not sediment MeHg production or a total system MeHg production (which would need a sediment-water incubation). By overstating the implications, you detract from the value of the truly novel findings – in my opinion that is the connection between methylation and CO₂ in the water column and the methanogen types (acetoclastic vs hydrogenic).

My recommendation is to reduce the claims of how this research shows worldwide reductions in lake methylation by 30 – 90% to something more grounded and reasonable: it is applicable to Hg-impacted, hypereutrophic lake water column methylation. These conditions are not very common, but understanding the mechanisms that you have elucidated is very helpful for the scientific community and will lead to follow-up research that will cite your paper and look more closely at scenarios that mimic real-world conditions. You can comment on how it might be applicable to non-Hg impacted and hypereutrophic lake water column methylation and if it is applicable to sediment methylation, but don't assume that it automatically is and be grounded in the theories of why it may be. You can also comment on the analytical challenges of running this type of experiment without adding Hg or AOM and how it could be accomplished in the future.

Sediment vs Water Column Methylation

Significantly more (> 10x) MeHg is produced in the sediment where mercury, microbial activity, and anoxia are all significantly higher. So even if water column methylation decreases by 80%, this does not necessarily mean a significant decrease in the total water column MeHg concentration. You briefly mentioned this in the supplemental material, but generally avoided the topic. I think it is a very significant stretch to say that increasing CO₂ decreases MeHg production in lakes as a whole. It is more powerful to be grounded in the implications of your data.

However, the compelling results are meaningful for water column methylation, which has been of increasing interest within the scientific community and understudied compared to sediment MeHg production. If masses are equal, water column methylmercury production has a much greater impact on human health due to its closer proximity to the pelagic food web. Your research is extremely helpful from this standpoint.

Grounding the results of the Mesocosm Experiment

In the main paper, it appears as if this is a natural mesocosm test and you have to read the supplemental material to find that you added 200 ng/L as HgCl₂. This is not a natural system – most eutrophic lakes have < 2 ng/L and > 20 ng/L is extremely rare. 200 ng/L in the water column only occurs in the most impacted lakes (this is 0.2 ug/L!). Your citation lists sediment and anaerobic digestion studies for the range of (20 – 1000 ng/L). The sediment study is a contaminated site and not natural.

Natural lake water generally has ~0.1 – 2 ng/L. If you were trying to extrapolate on natural conditions then you would have just used normal lake water, which already has some Hg in it and adding 1 ng/L might be reasonable.

However, I do understand the analytical difficulties of measuring MeHg when it is below 0.5 ng/L (which would be the outcome when water column Hg is generally < 1 ng/L) and I understand that it is just not feasible.

So it's not a deal breaker and it is still informative, but it's important not to overstate the implications of this if you didn't test the methylation changes without adding Hg. So it is important to be upfront that the mesocosms were run in absolute worst case scenario conditions and be very blunt about stating that these results are applicable to lakes that have water column mercury concentrations > 100 ng/L – this is almost exclusively contaminated sites and not natural sites.

Need to demonstrate that you did not enhance anaerobic conditions by adding CO₂

In the experimental design, it appears as if you may have introduced more oxygen into the controls than the treatments by aerating the incubator with air - this keeps the incubator atmosphere at 20% oxygen saturation. For your treatments, only adding pure CO₂ would purge the oxygen and not replenish it, allowing it to get lower in the incubator and ultimately lower in the tubes. Please make the methodology section more robust to explain how you overcame the potential change in the incubator oxygen concentrations - if you measured oxygen in the incubator or oxygen/redox in the tubes, this would suffice.

Model equation

The equation below results in 31.1% decrease in methylation if no DOC or MeHg are present. I think this multiple linear regression needs to be forced through the origin (set b = 0). This model is also biased towards a reduction, you don't have to calculate anything to know that it will result in a significant decrease in methylation under all circumstances. This exaggerates the true impact of your research, which does provide prominent evidence of decreasing water column

methylation.

The data that was used to produce the multiple linear regression also needs to be shown, it is more helpful than supplemental table 1, which is helpful, but just lit review data. The only way in which a model should always produce a decrease in MeHg is if every single mesocosm experiment resulted in a decrease. If even one out of the 45 experiments showed a decrease, then this would need to be incorporated into the model. I can't see the data so I don't know.

Another issue I have is that your results have 5-30 ng/L MeHg as the outcome in the non-elevated CO₂ scenarios, but the data set you used in your model has < 1 ng/L for more than 200 lakes from you lit review, with the average being ~0.1 ng/L. So I don't think it is appropriate to use a model where the MeHg is > 50x more than the systems you are modeling. I would strongly reconsider the parameters that affect the model - DOC seems appropriate, but are there other factors like pH, conductivity, redox, sulfate, etc?

R (Rate of inhibited MeHg (%) by elevated CO₂) = $(0.020 \pm 0.003) \times [\text{DOC}] + (0.013 \pm 0.002) \times [\text{MeHg}] + (0.311 \pm 0.03)$,
 $r^2_{\text{adjust}} = 0.753$, $p < 0.001$

Novelty of your research

Your findings that increasing CO₂ shifted the methanogen community from acetoclastic (more significant methylators) to hydrogenotrophic (less methylation capabilities) was very compelling and helpful for the scientific community. I think you should comment on the energetics and biochemistry of hydrogenotrophic vs acetoclastic metabolism more. Hydrogenotrophic metabolism produces more energy per mole of methane produced and is preferred if all substrates are equal. However, the law of the limiting factor suggests that it is much more complex than simply carbon dioxide concentration – which should not often be limited.

Note that 4 hydrogen molecules are needed per molecule of carbon dioxide. In anaerobic respiration, approximately 2 moles of H₂ are produced per mole of acetate. So as AOM is broken down, hydrogen will be limited compared to acetate and carbon dioxide and hydrogen production would need to be increased in order to stimulate hydrogenotrophic metabolism. I suspect fermenting organisms that produce more hydrogen than acetate were involved, since the incubations were performed in the dark and because you aerated the control (which introduces oxygen), but only added CO₂ to the treatments – they may not have had enough oxygen for redox compound cycling (Fe³⁺ → Fe²⁺ → Fe³⁺; SO₄ → H₂S → SO₄, etc). There are also algae, cyanobacteria, and green/purple sulfur bacteria that produce hydrogen during photosynthesis. Since they would be stimulated by carbon dioxide (especially in 2 month incubation), they could increase the hydrogen concentration needed to encourage hydrogenotrophic growth and although your experiment didn't look at this, it is a reasonable connection to your results - increasing CO₂ would encourage them and increase hydrogen and carbon dioxide, ultimately selecting for hydrogenotrophic metabolism.

Minor Comments

Figure 2 – amazing graph, but it's difficult to interpret that it is ~90% reduction due to the way it is shown and the label. I would suggest having the Y axis be 0% at the top and -100% at the bottom, with the larger bars further downwards. This emphasized that CO₂ had a negative impact on methylation. The Y axis label is kind of confusing, I would suggest "MeHg production compared to the control" and then the negative numbers will highlight that increased CO₂ led to decreased MeHg production.

Line 340 – change to "collected using a plankton net and stored at 4 C"

Line 347 – change experiment to experiments

Line 351 – add "of" inbetween "the mechanisms of elevated Co₂-impacted"

Line 363 – is this supposed to be 20 ml?

Line 386 – "the" is typed twice before observed, remove one.

Supplemental Material

Fig T3-2 – adjust the X-axis so that it starts at 0.0. It is slightly off and starts < 0.0.

Version 1:

Reviewer comments:

Reviewer #1

(Remarks to the Author)

This paper considers the question of how a warmer climate with higher atmospheric CO₂ concentrations and algal blooms will affect the concentrations and net production of neurotoxic methylmercury in lakes. This is a relevant question, and the paper presents an ambitious, well-executed set of experiments to answer that question. The authors have responded very well to an earlier round of reviews with new experiments and revisions to address the first set of reviews. I think this manuscript could be a valuable addition to the literature that is appropriate to the readership of Nature Communications. I have just two minor concerns that I suggest the authors' address.

The first concern is that the paper claims that the findings show that elevated CO₂ will counterbalance the effect of algal blooms on MeHg concentrations in lake, and will help stabilize MeHg levels. I think this suggestion of stability in future MeHg levels is claiming somewhat more than is possible given the complexity of mercury methylations. Claiming a little less

would be more convincing in my ears. For instance, in the abstract (line 57), to write “counteract” rather than “counterbalance” would be more appropriate. I also don’t think it is appropriate to speak of ‘helping to stabilize future MeHg concentrations’ (paraphrasing lines 59-60). To speak about counteracting is quite important enough. The other minor concern is that I tried to look for what the effect of adding 1000 ug/L of algal organic matter without changing the CO₂ levels was in the experiments. That information may be there, but I think this effect deserves to be more clearly stated in the text (apologies if I missed that). I could locate in diagrams the experimental effect of increasing CO₂ without increasing algal AOM, but those results would also be worth making clear, as a complement to the focus on the effect of combining increases in algal organic matter and CO₂.

Reviewer #2

(Remarks to the Author)

The images or other third party material in this Peer Review File are included in the article’s Creative Commons license, unless indicated otherwise in a credit line to the material. If material is not included in the article’s Creative Commons license and your intended use is not permitted by statutory regulation or exceeds the permitted use, you will need to obtain permission directly from the copyright holder.

Response to reviewers

For clarity, we present reviewer comments in *italics*, our response to comments in regular text, and a description of the changes to the manuscript in underlined text. Revisions in the revised manuscript and supplementary information are highlighted in **green**.

Reviewer #1:

Lei et al. present a detailed study on the relationship between atmospheric CO₂ and methylmercury production in freshwater lakes. It is a novel and interesting work that shows a possible pathway of reduced methylmercury under global warming, especially related to the decrease in hgcA genes. However, we have some concerns about its methods and arguments regarding their key findings on inhibited MeHg production and projected reduced MeHg levels in global lakes.

Response: We sincerely appreciate your thorough and insightful feedback, which has greatly improved scientific rigor and clarity of our study. In response to your concerns on **methods** and **result interpretation**, we have conducted **NEW** experiments and implemented corresponding revisions in the revised manuscript. Our detailed responses to each comment are provided below.

Major concerns:

The implications of the experimental design with no-light and constant temperature conditions need to be considered in the model extrapolation the results to lakes globally. “All vials were incubated in the dark at 25°C to mimic the prevalent degradation conditions in lakes and preclude MeHg photodegradation” (supplementary information L326-328). To project MeHg dynamics in global lakes, the influence of light, but especially temperature changes need to be mentioned. A significant rise in CO₂ concentration will drive a marked increase in temperature. How do you exclude the potential significant temperature influence on methylation and demethylation activities in global lakes when estimating the reduction in MeHg levels caused by elevated CO₂? High temperature can likely stimulate microbial activities, including Hg methylation. The paper could still be published with the existing data and model, but the implications of not considering light and temperature need mentioned.

Response: We appreciate your insightful comments regarding the implications of **light** and **temperature** for extrapolating our results to global lakes. These factors could be important in affecting MeHg dynamics in aquatic systems, and

we address their relevance as follows:

(1) Temperature: We agree with you that elevated atmospheric CO₂, often coupled with global warming, may jointly regulate microbial activity, thereby influencing both Hg methylation and demethylation processes ¹. To examine this interplay, we conducted **NEW** experiments simulating future climate scenarios with **elevated CO₂ AND/OR increased temperature**:

1) “Control”: 25°C incubation under ambient CO₂ (420 ppm).

2) “Warming”: 29.4°C incubation ($\Delta T=4.4^\circ\text{C}$), consistent with the SSP5-8.5 projections for 2100 from IPCC AR6 ² under ambient CO₂ (420 ppm).

3) “Elevated CO₂”: 25°C incubation under elevated CO₂ (1000 ppm).

4) “Warming × Elevated CO₂”: 29.4°C incubation under elevated CO₂ (1000 ppm).

Results showed that warming alone increased net MeHg production by 17–37% ($p < 0.01$) in both control lake water and algal biomass–amended systems, likely due to enhanced microbial activity ³. In contrast, elevated CO₂, either alone or in combination with warming, significantly suppressed net MeHg production by 26–74% relative to the control ($p < 0.001$). Importantly, the inhibition rates under the “**Elevated CO₂**” and “**Warming × Elevated CO₂**” treatments (33–76% reduction) were **not significantly different** ($p > 0.05$; Supplementary Fig. 5). Thus, under elevated CO₂ conditions, temperature increases had no significant additional effect on microbial MeHg production—regardless of whether natural lake water or eutrophic (algae-amended) conditions were considered. This demonstrates that the **inhibitory effect of elevated CO₂ on microbial Hg methylation outweighs the stimulatory effect of warming**. Consequently, our model remains valid for projecting global-scale trends in MeHg production under climate change, with CO₂ elevation as the dominant driver rather than temperature rise.

Supplementary Fig. 5 Effects of warming and elevated CO₂ on MeHg production. “Lake water”: unfiltered lake water from Yangshan Lake (Nanjing City, China); “+ 1000 Algae”: unfiltered lake water (40 mL) amended with 0.005 g of algal biomass. “Control”: incubated at 25°C under ambient CO₂ (420 ppm) for 7 days. “Warming”: incubated at 29.4°C ($\Delta T=4.4^\circ\text{C}$) and ambient CO₂ (420 ppm). “Elevated CO₂”: incubated at 25°C under elevated CO₂ (1000 ppm); “Warming × Elevated CO₂”: incubated at 29.4°C under elevated CO₂ (1000 ppm). Prior to incubation, 200 ng·L⁻¹ of HgCl₂ was added into the lake water. Different lowercase letters above the bars indicate significant differences among treatments ($p < 0.05$).

For clarity, we have added the following discussion in the main text and supplementary text.

Main text (Page 10, Line 179-184): “Moreover, under combined climate forcing (elevated CO₂ and/or warming, $\Delta T=4.4^\circ\text{C}$, SSP5-8.5 by 2100), MeHg production inhibition under “Elevated CO₂” and “Warming × Elevated CO₂” treatments (33–76% reduction) did not differ significantly ($p > 0.05$; Supplementary Fig. 5), indicating that the inhibitory effect of elevated CO₂ outweighs the stimulatory impact of warming, thus supporting the robustness of our model in projecting global MeHg trends.”

Supplementary Text 1 (Page S6, Line 107-121): “Furthermore, elevated atmospheric CO₂ typically co-occurs with global warming, which may alter microbial processes involved in Hg methylation and demethylation¹. To clarify this interplay, we conducted supplementary experiments simulating different climate scenarios: “Control” (25°C, ambient 420 ppm CO₂), “Warming” (29.4°C,

$\Delta T=4.4^{\circ}\text{C}$, consistent with SSP5-8.5 projections for 2100, ambient 420 ppm CO_2), “Elevated CO_2 ” (25°C , elevated CO_2 of 1000 ppm), and “Warming × Elevated CO_2 ” (29.4°C , elevated CO_2 of 1000 ppm). Results showed that warming alone increased net MeHg production by 17–37% ($p < 0.01$), while elevated CO_2 , either alone or combined with warming, significantly suppressed MeHg production (26–74% reduction vs. Control) with comparable inhibition rates (33–76%, $p > 0.05$, Supplementary Fig. 5). These findings indicate that the inhibitory effect of elevated CO_2 on microbial Hg methylation outweighs the stimulatory effect of rising temperature, supporting the robustness of our model in projecting global MeHg trends despite concurrent warming.”

(2) Light: Light is a key regulator of MeHg dynamics—particularly via photodegradation⁴, but its effect is strongly modulated by eutrophication status, which was central to our experimental design. Previous studies have shown that algal organic matter (AOM) from eutrophic conditions significantly inhibits MeHg photodegradation through light-shielding effects⁵. To specifically isolate the impact of elevated CO_2 on microbial Hg methylation—the focus of this study—we incubated all samples under dark conditions at 25°C to eliminate photodegradation interference. This setup is particularly relevant to eutrophic lake systems, where high AOM concentrations **substantially inhibited** light penetration and photodegradation.

Importantly, during incubation, DOM concentrations showed no significant differences (except at day 60) between ambient and elevated CO_2 treatments ($p > 0.05$, Supplementary Fig. T1-1). Thus, while MeHg concentrations would theoretically be lower under light exposure than in dark incubation, **the relative reduction rates are expected to be similar under ambient (420 ppm) or elevated CO_2 (1000 ppm) CO_2 levels**, because DOC levels were not significantly altered by elevated CO_2 . Consequently, the observed inhibitory effect of elevated CO_2 on net MeHg production remains valid even when photodegradation is considered, supporting the robustness of our model extrapolation. Therefore, **light** effects were **NOT** considered in our global extrapolation. Future studies should integrate *in-situ* light regimes and the long-term effect of elevated CO_2 on water DOC level to capture broader lake types.

Supplementary Fig. T1-1 Dynamic changes of DOC levels during incubation under ambient CO₂ (420 ppm) or elevated CO₂ (1000 ppm) levels.

To address your concerns, we have incorporated these points in our model uncertainty analysis (See Supplementary Text 1, Page S3-S4, Line 51-65): “First, we quantified the responses of microbial Hg methylation to elevated CO₂ levels over a two-month incubation under dark conditions. This design intentionally excluded photodegradation, which is strongly suppressed in eutrophic lakes due to the light-shading effect of AOM⁵, but remains important in oligotrophic or clear-water systems. While MeHg concentrations would be lower under light exposure compared to dark incubation, the relative reduction rates are expected to remain similar under both ambient (420 ppm) and elevated (1000 ppm) CO₂, because DOC concentrations were not significantly altered by CO₂ elevation ($p > 0.05$, Supplementary Fig. T1-1). Thus, the observed inhibitory effect of elevated CO₂ on net MeHg production is robust even when photodegradation is considered. Future work should incorporate *in-situ* light regimes to extend applicability to a broader range of lake types. In addition, further research is needed to assess long-term responses, as Hg-methylating microbes such as methanogens may gradually acclimate to elevated CO₂ and other environmental changes during prolonged incubation^{6, 7}.”

Furthermore, the global model on MeHg in lake water only uses DOC and MeHg as the parameters. The experiment, however, focuses on Algal Organic Matter (AOM), but DOC is used by the authors use to model the global trend of MeHg levels. The fact that AOM is only a fraction of DOC, and DOC is more complex regarding methylation and demethylation of Hg also needs to be acknowledged as a limitation of the model.

Response: We thank you for highlighting the distinction between dissolved organic carbon (DOC) and algal organic matter (AOM), as well as its

implications for our global model. This distinction is indeed critical, since although AOM can constitute a major fraction of DOC in eutrophic lakes, it coexists with diverse autochthonous and allochthonous sources ⁸. Such compositional heterogeneity exerts differential controls on Hg methylation and demethylation processes ⁹.

Our model employs DOC as the parameter primarily because global-scale DOC datasets are available, enabling broad extrapolation. However, we acknowledge that DOC is heterogeneous, with AOM (algae-derived, autochthonous) representing a functionally distinct fraction—particularly in eutrophic lakes, where it dominates and strongly stimulates methanogen-mediated Hg methylation ¹⁰. In contrast, allochthonous DOC (e.g., soil-derived organic matter, SOM) generally exerts weaker or even different effects on Hg transformation processes ⁹.

To directly address your concern, we performed **NEW** experiments using **mixtures of autochthonous and allochthonous DOM**, to clarify how varying AOM contributions (within the broader DOC pool) affect CO₂-driven MeHg inhibition. We established a gradient of DOM mixtures representing different autochthonous: allochthonous ratios (100:0, 80:20, 50:50, 20:80, and 0:100), corresponding to AOM (autochthonous), M-1, M-2, M-3, and SOM (allochthonous), respectively. All treatments were buffered to neutral pH (≈ 7) to avoid pH effects. The results confirmed that AOM is a much stronger driver of methylation than SOM, consistent with previous studies ^{9, 10}. Importantly, elevated CO₂ (1000 ppm) significantly inhibited net MeHg production only when AOM made up $\geq 50\%$ of the DOM pool ($p < 0.001$). The inhibition rate declined progressively from 75% (pure AOM) to 68% (80:20, M-1), 52% (50:50, M-2), and became statistically insignificant ($< 10\%$) when AOM constituted $< 50\%$ of the DOM pool (20:80 of M-3 and pure SOM; Supplementary Fig. 4).

These results indicate that our global model—which implicitly equates “high DOC” with “AOM dominance”—may **overestimate** CO₂-driven MeHg suppression in lakes where terrestrial DOM dominates. Nevertheless, the prediction error for eutrophic lakes (where AOM is typically dominant and where our study is primarily focused as recognized MeHg hotspots ¹⁰) remains moderate, with an overestimation of 10–44% (mean $28\% \pm 24\%$). To reduce this bias, we normalized the inhibition rates of net MeHg production in the “Algae” treatments using the result from a mixture of autochthonous and allochthonous organic matter (AOM: SOM =80%: 20%). This **refinement** slightly narrowed the global projected inhibition range: from 31–86% (mean 50%) before normalization to 33–74% (mean 47%) after normalization.

Supplementary Fig. 4 Effects of elevated CO₂ (1000 ppm) on MeHg production compared to ambient CO₂ (420 ppm) under different DOM sources and mixing ratios.

Soil samples were collected from the riparian zone of Chaohu Lake. DOM was extracted from soil and algal biomass using water (soil: water = 1:2, algae: water = 1:100, w/w) in the dark at 25 °C for 12 h, centrifuged (4000 rpm, 10 min), filtered (0.45 μm PES), and diluted to 20 mg·L⁻¹ DOC. To generate mixing gradients, algae- (AOM) and soil-derived (SOM) DOM stock solutions were combined at fixed ratios: 100:0 (AOM), 80:20 (M-1), 50:50 (M-2), 20:80 (M-3), and 0:100 (SOM). For the “Lake water”: 40 mL of unfiltered lake water; for each “+DOM”: 20 mL lake water + 20 mL DOM solution (final DOC ≈ 12.5 mg·L⁻¹, comparable to the “+1000 Algae” treatment ≈ 13.3 mg·L⁻¹). All treatments were incubated for 7 days under either ambient (420 ppm) or elevated (1000 ppm) CO₂. The asterisk above the bars indicates a significant difference between “420 ppm CO₂” and “1000 ppm CO₂” (*: $p < 0.05$; **: $p < 0.01$; ***: $p < 0.001$; without *: $p > 0.05$).

For clarity, we have added the following discussion in the main text and supplementary text to explicitly acknowledge this limitation and to incorporate our new DOM-mixing results.

Main text (Page 10, Line 173-179): “We also established DOM gradients reflecting different autochthonous: allochthonous ratios (100:0 to 0:100; AOM vs. Soil organic matter, SOM). Results confirmed that AOM was a stronger driver of methylation than SOM, consistent with previous studies^{9,10}, and that elevated CO₂ significantly inhibited net MeHg production only when AOM comprised ≥50% of the DOM pool ($p < 0.001$; Supplementary Fig. 4), underscoring the need to clarify how DOM composition modulates CO₂-driven

MeHg inhibition.”

Supplementary Text 1 (Page S5-S6, Line 89-106): “Our model incorporates DOC as a parameter primarily due to the availability of global DOC datasets, which enable broad-scale extrapolation. However, DOC represents a heterogeneous pool encompassing both autochthonous and allochthonous fractions that may differentially regulate Hg methylation and demethylation ⁹. Specifically, algae-derived AOM, particularly in eutrophic lakes, is a functionally distinct driver of microbial Hg methylation ¹⁰, while soil-derived SOM typically exerts weaker effects ⁹. To clarify how AOM contributions modulate CO₂-driven inhibition of MeHg production, we conducted additional experiments with mixed DOM sources: pure AOM, pure SOM, and AOM:SOM ratios of 80:20, 50:50, and 20:80. Elevated CO₂ significantly suppressed net MeHg production only when AOM constituted $\geq 50\%$ of the DOM pool ($p < 0.001$). The inhibition rate declined from 75% (pure AOM) to 68% (80:20), 52% (50:50), and became negligible (<10%) when AOM was <50% (Supplementary Fig. 4). These findings demonstrate that while our model may overestimate MeHg declines in lakes dominated by terrestrial DOM, the deviation for eutrophic lakes (the principal focus of this study) is moderate. After normalizing by the AOM:SOM ratio, projected global inhibition ranges were refined from 31–86% (mean 50%) to 33–74% (mean 47%).”

Finally, the concluding lines of the paper state that its findings “offer a potential explanation for this mismatch. I think that statement is incorrect. The mismatch, as the authors present it, is that the reductions in emissions have only resulted in a decrease in human exposure that is proportionally smaller than emission reductions. The findings of the present paper suggest that even without any change in emissions, the MeHg in water would be going down as CO₂ levels rise, thus reducing human exposure even if Hg emissions were not reduced. This would tend to make human exposure decline faster than Hg reductions. The authors should reconsider what they claim for this paper in regards to explaining the current mismatch between Hg emission reductions and human exposure.

Response: We sincerely thank you for your insightful comment, which has helped us identify an inaccuracy in our original conclusion. We fully agree that our findings do not directly explain the observed mismatch between reductions in Hg emissions and the proportionally smaller declines in human MeHg exposure. In response, we have revised the conclusion to provide a more accurate and nuanced interpretation that better reflects our experimental evidence (Page 20-21, Line 377-394):

“Per unit reduction in Hg emissions, human exposure has been estimated to decline by only one-third to one-half, reflecting the influence of complex processes such as legacy Hg in environmental reservoirs and bioaccumulation lags in food chains. This observed “mismatch”—where exposure decreases less than expected from emission reductions—likely arises from such multi-faceted drivers, including remobilization of legacy Hg by natural and anthropogenic factors ¹¹.

Our study identifies a distinct climatic pathway that further complicates this emission–exposure relationship: elevated CO₂ suppresses microbial MeHg production in lakes, leading to reduced aquatic MeHg levels regardless of emission changes. This pathway would amplify the decline in human exposure beyond direct emission reductions, and should be accounted for when evaluating the effectiveness of global Hg mitigation efforts and interpreting temporal trends in biotic MeHg concentrations. Thus, our findings do not resolve the current mismatch, but rather reveal a previously unrecognized climate–microbe–MeHg interaction that may shape future exposure risks. Specifically, future increases in atmospheric CO₂ may counteract warming-driven enhancements of algal blooms, helping to stabilize aquatic MeHg levels under climate change.”

Minor comments:

1. *The abstract section on hydrogenotrophic and acetoclastic methanogens does not talk about differences in the hgcA genes between the two types of methanogens. It would be clearer if it did.*

Response: We appreciate this valuable suggestion, which prompts us to clarify the difference in *hgcA* gene distribution between hydrogenotrophic and acetoclastic methanogens. Based on our findings, we have revised the relevant description to explicitly link metabolic types to their Hg methylation potential via *hgcA* (Page 3, Line 51-55): “These shifts result from CO₂-induced changes in carbon substrates, promoting hydrogenotrophic methanogens (e.g., *Methanobacterium*)—which typically lack the *hgcA* methylation gene—over acetoclastic strain (e.g., *Methanosarcina*) that harbor *hgcA* and mediate Hg methylation. This competitive shift thus reduces Hg methylation by methanogens.”

2. *The North arrow needs to be indicated on the map Figure 1A.*

Response: Added as suggested in Figure 1A.

Fig. 1 Elevated CO₂ inhibited MeHg production in lakes along the middle and lower reaches of the Yangtze River in China. (a) “Lake water”: unfiltered lake water. (b) “+1000 Algae”: unfiltered lake water amended with fresh algal biomass to simulate bloom (resulting in approximately 1000 µg·L⁻¹ chlorophyll a) in Experiment A. Inhibited MeHg production rates by elevated CO₂ are denoted as the percentage changes of MeHg levels between “Ambient CO₂” and “Elevated CO₂”.

3. Only data from Chaohu and Yangshan Lakes are presented in Figure 2. In supplementary Figure 2, Xuanwu Lake data is added to demonstrate the negligible impact of elevated CO₂ levels on demethylation. How about other lake data? Are they available in the supplementary information?

Response: We thank you for your question regarding data presentation. To clarify, our full dataset includes results from **45 freshwater lakes spanning ~1,200 km along the Yangtze River**. In the main text, we highlighted three representative lakes—Chaohu, Yangshan, and Xuanwu—to illustrate key trends under contrasting ecological contexts. © **Chaohu Lake**, a large

eutrophic system with frequent algal blooms, represents conditions of high AOM where methanogen-mediated methylation is most active ¹⁰. © **Yangshan Lake** (suburban, relatively undisturbed) and **Xuanwu Lake** (urban, anthropogenically impacted) capture contrasting trophic states and land-use settings ¹². This selection allows us to demonstrate the consistency of CO₂-induced effects across diverse lake types.

The **core finding**—a ubiquitous reduction in net MeHg production under elevated CO₂ (14–96% across all 45 lakes)—is supported by the full dataset. Aggregated results are provided in Supplementary Fig. 1, confirming that the patterns seen in the three highlighted lakes are **representative** of the broader set. Regarding **demethylation**, Supplementary Fig. 7 presents results from Chaohu Lake showed Hg methylation rates (k_m) were 83%–89% lower under elevated CO₂ than ambient conditions in both natural or eutrophic waters ($p < 0.001$), whereas MeHg demethylation rates (k_d) did not differ significantly between treatments, reinforcing our conclusion that the observed MeHg declines are primarily driven by inhibited methylation rather than altered demethylation.

To make this clearer, we have added explanatory text in Supplementary Text 9 (Page S30-S31, Lines 517–524):

“In Experiments B–H, water from Chaohu and Yangshan Lakes was selected to represent a range of ecological contexts along the Yangtze River. Chaohu Lake, a large eutrophic system characterized by frequent algal blooms, exemplifies high AOM conditions and active methanogen-mediated methylation. Yangshan Lake, a suburban lake with relatively low human disturbance, was chosen to capture contrasting trophic states. While the two lakes are highlighted for clarity, the conclusions are supported by data from 45 freshwater lakes across the Yangtze Basin, ensuring broad applicability.”

4. In Figure 2, only %MeHg data is shown. How about the MeHg concentrations?

Response: As suggested, we have supplemented the MeHg concentrations in different scenarios under ambient (420 ppm) or elevated CO₂ (1000 ppm) conditions (Supplementary Fig. 2).

Supplementary Fig. 2 Net MeHg production in different scenarios under ambient (420 ppm) or elevated CO₂ (1000 ppm) conditions. (a) Different CO₂ levels; (b) Different trophic statuses; (c) Different decomposition stages. “Lake water”: unfiltered lake water. “+200 Algae”, “+1000 Algae”, and “+2000 Algae”: unfiltered lake water (40 mL) amended with 0.001, 0.005, and 0.01 g of algal biomass, respectively. Before sampling on “Day 0”, all lake water was equilibrated for 4 h with Hg(II). Different lowercase letters above the bars indicate significant differences among different treatments ($p < 0.05$). The asterisk above the bars indicates a significant difference between “420 ppm CO₂” and “1000 ppm CO₂” (*: $p < 0.05$; **: $p < 0.01$; ***: $p < 0.001$; without *: $p > 0.05$).

Reviewer #2:

Response: We sincerely appreciate your thorough and insightful feedback, which has significantly strengthened both the scientific rigor and clarity of our manuscript.

Reviewer #3:

I am very impressed by this extensive research study and the thorough effort to address potentially confounding factors in the supplemental material (demethylation, FeRB/SRB, etc). There was a lot of great research performed and a lot of novel findings. However, I believe that the context of the publication needs to be reconsidered – this is water column methylmercury production and not sediment MeHg production or a total system MeHg production (which would need a sediment-water incubation). By overstating the implications, you detract from the value of the truly novel findings – in my opinion that is the connection between methylation and CO₂ in the water column and the methanogen types (acetoclastic vs hydrogenic).

My recommendation is to reduce the claims of how this research shows worldwide reductions in lake methylation by 30 – 90% to something more grounded and reasonable: it is applicable to Hg-impacted, hypereutrophic lake water column methylation. These conditions are not very common, but understanding the mechanisms that you have elucidated is very helpful for the scientific community and will lead to follow-up research that will cite your paper and look more closely at scenarios that mimic real-world conditions. You can comment on how it might be applicable to non-Hg impacted and hypereutrophic lake water column methylation and if it is applicable to sediment methylation, but don't assume that it automatically is and be grounded in the theories of why it may be. You can also comment on the analytical challenges of running this type of experiment without adding Hg or AOM and how it could be accomplished in the future.

Response : We sincerely appreciate your time and insightful comments. Particularly, your **actionable suggestions** are valuable in enhancing our study, which have prompted **additional experiments** to further enhance the scientific rigor and overall quality of our manuscript. We have addressed all your concerns—both the general comments above and specific points below—in the following detailed responses.

Sediment vs Water Column Methylation

Significantly more (> 10x) MeHg is produced in the sediment where mercury, microbial activity, and anoxia are all significantly higher. So even if water column methylation decreases by 80%, this does not necessarily mean a significant decrease in the total water column MeHg concentration. You briefly mentioned this in the supplemental material, but generally avoided the topic. I think it is a very significant stretch to say that increasing CO₂ decreases MeHg production in lakes as a whole. It is more powerful to be grounded in the implications of

your data.

However, the compelling results are meaningful for water column methylation, which has been of increasing interest within the scientific community and understudied compared to sediment MeHg production. If masses are equal, water column methylmercury production has a much greater impact on human health due to its closer proximity to the pelagic food web. Your research is extremely helpful from this standpoint.

Response: We appreciate your insightful comments highlighting the significance of sedimentary MeHg production and its implications for our conclusions. Your suggestions underscore the importance of clarifying the scope of our study while carefully contextualizing it within the broader framework of aquatic Hg cycling.

Our study specifically focuses on water column MeHg production, because this pool directly interfaces with pelagic food webs—where MeHg enters phytoplankton, zooplankton, fish, and ultimately human diets¹³. Water column MeHg is therefore more tightly linked to pelagic trophic transfer, making its regulation particularly critical for understanding human exposure risks¹⁴. For instance, across a latitudinal field survey of 12 subtidal sites within three estuaries, Taylor et al. (2019) found that MeHg concentrations in resident fish were linearly predicted by dissolved water-column MeHg (Atlantic silverside $r^2 = 0.73\text{--}0.82$, $p < 0.01$), while **NO** significant relationship was observed with bulk sediment MeHg ($p > 0.05$)¹⁵. This finding illustrates why the study of methylation processes in water column, rather than sedimentary pathways, remains important for predicting MeHg bioaccumulation in edible aquatic organisms.

To further address your concern regarding sediments, our recent parallel study on coastal marine sediments (East China Sea) reveals that elevated CO₂ (1000 ppm) suppressed sedimentary MeHg production by 54–77% relative to ambient conditions (420 ppm)¹⁶. This inhibition was associated with a substantial decrease (67–99%) in the abundance of the Deltaproteobacteria-*hgcA* gene, the key functional gene for Hg-methylating microbes in marine sediments. The consistency of this suppression across both water column and sediment systems suggests that elevated CO₂-induced inhibition of microbial Hg methylation may represent a generalizable pattern in aquatic environments, thereby reinforcing the broader significance of our findings.

Nevertheless, it is important to note that although sediments are the dominant zones of MeHg production, the partitioning of MeHg between sediment solids and overlying waters strongly constrains the flux of sediment-derived MeHg to the water column. This partitioning is described by a partition

coefficient (K_d), which ranges from 10^3 – 10^5 L·kg⁻¹¹⁷. Using an average K_d value of 10^4 L·kg⁻¹ and typical sedimentary MeHg concentrations of 0.1–0.5 ng·g⁻¹ in freshwater lakes¹⁸, we estimate that elevated CO₂-inhibited sedimentary MeHg (54–77% reduction) would contribute only 0.006–0.03 ng·L⁻¹ (average 0.018 ng·L⁻¹) to the overlying water. By comparison, our observed background MeHg concentrations in lake water were 0.04–0.47 ng·L⁻¹ (average 0.15 ± 0.10 ng·L⁻¹), indicating that **CO₂-impacted sediment-derived MeHg represents <12%** of the total aqueous MeHg pool. This aligns with previous findings (Li and Cai, 2013)¹⁹, which showed that only a small proportion of sedimentary MeHg is transported to the water column. Thus, while sedimentary processes remain critical for whole-lake budgets, the influence of reduced sedimentary methylation under elevated CO₂ on dissolved MeHg concentrations in the water column is limited. For this reason, our primary emphasis remains on *in situ* water-column MeHg production.

Furthermore, our water-column focus is justified by the fact that atmospheric CO₂ directly influences surface waters, where rapid gas exchange equilibrates CO₂ concentrations²⁰. In contrast, sedimentary CO₂ dynamics are moderated by porewater diffusion and *in situ* microbial processes, which buffer against direct atmospheric forcing. Thus, constraining our analysis to the water column enables a clearer isolation of atmospheric CO₂ effects on methylation—a process that most directly impacts human exposure through freshwater fisheries.

We acknowledge, however, that integrating sedimentary pathways is essential for a holistic understanding of lake-wide MeHg cycling. Future studies should therefore investigate how rising atmospheric CO₂ indirectly affects sedimentary methylation—for instance, through altered organic matter export from surface waters or porewater geochemical changes—to provide a more complete assessment of climate-driven impacts on lake MeHg budgets.

To directly address your suggestions and strengthen the rigor of our study, we have expanded our discussion (Supplementary Text 1, Page S4–S5, Line 70–88) to include: (1) the rationale for focusing on water-column processes, and (2) the limitations and uncertainties of not incorporating sedimentary processes. Specifically, we state:

“Another key factor that may interfere with the CO₂-MeHg nexus in lake waters is sediments, which represent the largest pool of Hg in aquatic ecosystems. Predicting climate-driven changes in freshwater MeHg levels also requires understanding how elevated CO₂ affects sedimentary MeHg production and water-sediment MeHg fluxes. Our recent parallel study on coastal marine sediments (East China Sea) reveals that elevated CO₂ (1000

ppm) also suppresses MeHg production in sediments, with reductions ranging from 54% to 77% compared to ambient conditions (420 ppm)¹⁶. However, considering the partitioning of MeHg between sedimentary solid and water phases (e.g., with partition coefficient K_d of 10^3 – 10^5 L·kg⁻¹¹⁷), the CO₂-impacted MeHg from sediment represents only a small fraction (e.g., <12%) of the total dissolved MeHg observed in lake water¹⁹. Nevertheless, this consistency across water column and sediment systems suggests that elevated CO₂-induced inhibition of microbial Hg methylation may represent a generalizable pattern across aquatic environments, reinforcing the broader relevance of our findings. Future studies will explore how elevated atmospheric CO₂ indirectly affects sedimentary methylation (e.g., via altered organic matter export from the water column or porewater chemistry) to better quantify total lake MeHg budgets under climate change.”

Grounding the results of the Mesocosm Experiment

In the main paper, it appears as if this is a natural mesocosm test and you have to read the supplemental material to find that you added 200 ng/L as HgCl₂. This is not a natural system – most eutrophic lakes have < 2 ng/L and > 20 ng/L is extremely rare. 200 ng/L in the water column only occurs in the most impacted lakes (this is 0.2 ug/L). Your citation lists sediment and anerobic digestion studies for the range of (20 – 1000 ng/L). The sediment study is a contaminated site and not natural. Natural lake water generally has ~0.1 – 2 ng/L. If you were trying to extrapolate on natural conditions then you would have just used normal lake water, which already has some Hg in it and adding 1 ng/L might be reasonable.

However, I do understand the analytical difficulties of measuring MeHg when it is below 0.5 ng/L (which would be the outcome when water column Hg is generally < 1 ng/L) and I understand that it is just not feasible.

So it's not a deal breaker and it is still informative, but it's important not to overstate the implications of this if you didn't test the methylation changes without adding Hg. So it is important to be upfront that the mesocosms where run in absolute worst case scenario conditions and be very blunt about stating that these results are applicable to lakes that have water column mercury concentrations > 100 ng/L – this is almost exclusively contaminated sites and not natural sites.

Response: We thank you for highlighting that our spiked inorganic mercury (IHg) concentration (200 ng·L⁻¹) exceeds typical lake levels. We acknowledge that this concentration does not represent pristine systems, where IHg is usually ~0.1–2 ng·L⁻¹, and instead reflects conditions characteristic of highly

contaminated sites. Here, we would like to clarify the rationale for this design and its implications for interpreting our results.

The 200 ng·L⁻¹ IHg spike was applied primarily to overcome analytical constraints. As you noted, quantifying net MeHg production in natural lake waters (where IHg is often <1 ng·L⁻¹ and MeHg <0.5 ng·L⁻¹ ²¹) is hampered by detection limits, making it difficult to capture subtle shifts induced by elevated CO₂. This design therefore prioritized detecting the microbial response to CO₂, which is the central focus of our study.

To address the concern about environmental relevance, we conducted **NEW experiments across a broad IHg gradient (2–500 ng·L⁻¹)**, encompassing both slightly contaminated and heavily impacted conditions ^{20, 22}. Although net MeHg production increased with higher IHg inputs, the inhibitory effect of elevated CO₂ (1000 ppm) was consistent across all treatments. Compared to ambient CO₂, net MeHg production was reduced by 21–32% in unamended lake water (mimicking low-trophic conditions) and by 66–84% in algal biomass-amended treatments (mimicking eutrophic conditions) (Supplementary Fig. 3). While a 200 ng·L⁻¹ spike better represents heavily contaminated rather than pristine systems, our gradient experiments confirm that the key mechanism—CO₂ reshaped methanogenic communities, particularly a decreased abundance of *hgcA*-containing acetoclastic methanogens—persists across environmentally relevant Hg gradients. Thus, although our experiments simulate lakes with elevated Hg levels (e.g., anthropogenically impacted systems), the identified CO₂-mediated pathway provides critical insights into methylation regulation even in low-Hg environments.

Supplementary Fig. 3 Effect of elevated CO₂ on MeHg production across different IHg concentrations. Levels equivalent to 2, 20, 100, 200 and 500 ng·L⁻¹ of HgCl₂ were spiked into the lake water, respectively. “Lake water”: unfiltered lake water from Yangshan Lake (located in Nanjing City, China); “+ 1000 Algae”: unfiltered lake water (40 mL) amended with 0.005 g of algal biomass from Chaohu Lake. The asterisk above the bars indicates a significant difference between “420 ppm CO₂” and “1000 ppm CO₂” (*: $p < 0.05$; **: $p < 0.01$; ***: $p < 0.001$; without *: $p > 0.05$).

To address your concerns, we have revised our discussion and included results of these **NEW** experiments to explicitly contextualize the results within this scope, avoiding potential overgeneralization and misleading (Page 10, Line 168-173): “The CO₂-driven suppression of microbial Hg methylation is independent of background Hg levels, as shown by spiking experiments across a broad inorganic Hg (IHg) gradient (2–500 ng L⁻¹). Elevated CO₂ consistently reduced net MeHg production relative to ambient CO₂ levels, with 21–32% inhibition in unamended lake water and 66–84% in algal biomass-added treatments (Supplementary Fig. 3).”

Need to demonstrate that you did not enhance anaerobic conditions by adding CO₂ in the experimental design, it appears as if you may have introduced more oxygen into the controls than the treatments by aerating the incubator with air - this keeps the incubator atmosphere at 20% oxygen saturation. For your

treatments, only adding pure CO₂ would purge the oxygen and not replenish it, allowing it to get lower in the incubator and ultimately lower in the tubes. Please make the methodology section more robust to explain how you overcame the potential change in the incubator oxygen concentrations - if you measured oxygen in the incubator or oxygen/redox in the tubes, this would suffice.

Response: We appreciate this critical concern regarding potential oxygen changes in our experiments, as maintaining consistent redox conditions is essential for examining and interpreting microbial Hg methylation responses to elevated CO₂. Here, we clarify the methodology and supplement relevant data to address this concern:

Our experimental design maintained **consistent** oxygen concentrations across all treatments to isolate CO₂ effects from potential anaerobiosis changes. Rather than using pure CO₂, we employed **gas mixtures** calibrated to target CO₂ levels (650 ppm and 1000 ppm) while maintaining atmospheric oxygen partial pressure (~20%). The gas mixtures were continuously bubbled into incubators at a constant flow rate (200 mL·min⁻¹), ensuring both CO₂ levels and oxygen concentrations were stable across control (420 ppm CO₂ in air) and treatments with elevated CO₂ throughout the incubation period. To verify this, we measured dissolved oxygen (DO) in triplicate culture tubes. The results demonstrated **NO** significant differences in dissolved oxygen (DO) levels between ambient (420 ppm) and elevated (1000 ppm) CO₂ treatments across most time points, consistently observed in both lake water controls and algal biomass-amended treatments ($p > 0.05$, Supplementary Fig. T8-1). Particularly, the measured DO levels in our experimental systems reflect the microaerophilic conditions characteristic of eutrophic lake water columns—under conditions where methanogens and other Hg-methylating microbes remain metabolically active.

Supplementary Fig. T8-1 Dissolved oxygen in “Lake water” (a) and “+1000 Algae” (b) during incubation under ambient CO₂ (420 ppm) or elevated CO₂ (1000 ppm) levels.

To avoid potential confusion, we have updated the Materials and Methods section to include: 1) explicit descriptions of the gas mixture protocols, and 2) DO monitoring data (Supplementary Fig. T8-1), thereby ensuring transparency about redox stability across treatments (Page S25, Line 402-409): “Meanwhile, elevated CO₂ levels used calibrated gas mixtures of air and commercially prepared CO₂ gas ensuring target CO₂ levels (approximately 650 or 1000 ppm) and atmospheric O₂ partial pressure (~20%). This design explicitly prevented CO₂-induced anaerobiosis shifts. Dissolved oxygen (DO) measurements confirmed no significant differences ($p > 0.05$) between ambient and elevated CO₂ treatments (1000 ppm) across lake water controls and algal-amended treatments at most time points (Supplementary Fig. T8-1), consistent with microaerophilic conditions in eutrophic lakes where methylators thrive.”

Model equation

The equation below results in 31.1% decrease in methylation if no DOC or MeHg are present. I think this multiple linear regression needs to be forced through the origin (set $b = 0$). This model is also biased towards a reduction, you don't have to calculate anything to know that it will result in a significant decrease in methylation under all circumstances. This exaggerates the true impact of your research, which does provide prominent evidence of decreasing water column methylation.

The data that was used to produce the multiple linear regression also needs to be shown, it is more helpful than supplemental table 1, which is helpful, but just lit review data. The only way in which a model should always produce a decrease in MeHg is if every single mesocosm experiment resulted in a decrease. If even one out of the 45 experiments showed a decrease, then this would need to be incorporated into the model. I can't see the data so I don't know.

Another issue I have is that your results have 5-30 ng/L MeHg as the outcome in the non-elevated CO₂ scenarios, but the data set you used in your model has < 1 ng/L for more than 200 lakes from you lit review, with the average being ~0.1 ng/L. So I don't think it is appropriate to use a model where the MeHg is > 50x more than the systems you are modeling. I would strongly reconsider the parameters that affect the model - DOC seems appropriate, but are there other factors like pH, conductivity, redox, sulfate, etc?

R (Rate of inhibited MeHg (%) by elevated CO₂) = $(0.020 \pm 0.003) \times [DOC] + (0.013 \pm 0.002) \times [MeHg] + (0.311 \pm 0.03)$, $r^2_{adjust} = 0.753$, $p < 0.001$

Response: Your constructive feedback on the linear regression model is greatly appreciated. We fully agree with you that limitations and uncertainties

exist in our model construction and extrapolation, and these should be clearly indicated in the manuscript to avoid potential misleading or overstating. To address these concerns, we have expanded our discussion of the methodology, dataset, and model limitations, with corresponding revisions incorporated throughout the manuscript.

(1) Linear regression: The regression equation R (Rate of inhibited MeHg (%)) by elevated CO_2 = $(0.020 \pm 0.003) \times [\text{DOC}] + (0.013 \pm 0.002) \times [\text{MeHg}] + (0.311 \pm 0.03)$, (adjusted $r^2 = 0.753$, $p < 0.001$) was derived from data spanning **45 lakes**, where all samples exhibited variable concentrations of DOC (3.2–28.7 $\text{mg}\cdot\text{L}^{-1}$) and MeHg (0.23–39.3 $\text{ng}\cdot\text{L}^{-1}$) (Supplementary Table 4). As noted, natural waters seldom, if ever, exhibit zero DOC or MeHg concentrations—DOC is ubiquitous in lakes due to organic matter inputs, and MeHg is consistently detected in lakes. Forcing the model through the origin (intercept = 0) leads to implausible results: at high DOC ($>20 \text{ mg}\cdot\text{L}^{-1}$), the predicted reduction exceeds 100% (e.g., $20 \text{ mg}\cdot\text{L}^{-1}$ DOC would yield 100.0% reduction under the forced model), which contradicts our experimental observations (maximum 96% reduction). Thus, retaining the intercept better reflects ecological realism, where baseline reductions (the intercept) may arise from CO_2 -induced microbial shifts independent of DOC/MeHg concentrations (e.g., direct effects on methanogen metabolism). Additionally, as requested by Reviewer #1, we found that the inhibitory effect of elevated CO_2 on MeHg production became more pronounced with increasing proportions of algal-derived organic matter (AOM) in the dissolved organic carbon (DOC) pool. To account for this effect, the inhibition rates of net MeHg production in the “Algae” treatments were normalized using the M1/AOM ratio (where M1 represents a mixture of 20% terrestrial organic matter and 80% AOM), thereby establishing a refined predictive model. This adjusted model was subsequently applied to extrapolate projected changes in MeHg concentrations across global lake systems. Before normalization, the estimated decreases in global lake MeHg levels ranged from 31% to 86% (mean: 50%). After normalization, the inferred inhibition range narrowed to 33–74% (mean: 47%).

(2) Dataset: The regression draws on **45 lake** microcosm experiments. In all these experiments, elevated CO_2 led to decreased net MeHg production, with reductions ranging from 14% to 96%. No lake experiment showed increased or unchanged MeHg production under elevated CO_2 , confirming the uniform inhibitory effect of CO_2 on net MeHg production. This consistency supports the model’s trend projection: The inhibition rates of net MeHg production varied across lakes, and these differences were mainly controlled by dissolved organic carbon (DOC) and MeHg concentrations (adjusted $r^2 =$

0.701). Specifically, higher ratios of DOC to MeHg correlated with stronger inhibition. This aligns with the role of AOM (a major DOC component)—AOM can enhance methanogen activity, thereby heightening the system's sensitivity to CO₂.

(3) Methylmercury concentration: We recognize that the MeHg concentrations in our microcosms (0.23–39.3 ng·L⁻¹) exceed typical lakewater levels reported in the literature (averagely 0.14 ng·L⁻¹, Supplementary Table 4). This discrepancy arises from experimental design: IHg amendments and algal additions (simulating eutrophication) were employed to enhance MeHg production, ensuring detectable MeHg levels in water. This approach was necessary because ambient lakewater MeHg levels (typically < 0.5 ng·L⁻¹ ²³) are insufficient for quantify CO₂'s effects on MeHg production. To validate our model, we compared simulated outputs with experimental data from 45 lakes. The model, when applied to 229 natural lake samples (DOC: 1.2–27.0 mg·L⁻¹; MeHg: 0.01–1.2 ng·L⁻¹), predicted MeHg concentration reductions of 34–79% (average 47%) under elevated CO₂. These predictions **align well** with our experimental observations (14–96%, mean = 63%), demonstrating strong model consistency.

(4) Additional parameters: Regarding additional parameters (pH, sulfate, etc.), our analysis focused on DOC and MeHg because: (1) DOC is a key determinant of methanogen activity in eutrophic lakes as AOM serves of carbon sources for methanogens ¹⁰; (2) Elevated CO₂-decreased abundance of Archaea-*hgcA* gene was significantly correlated with elevated CO₂-decreased MeHg production ($R^2=0.866$, $p<0.001$, Supplementary Fig. 8). Although factors such as sulfate may influence Hg methylation in certain environments, our results in the examined lakes revealed no significant associations between these variables and CO₂-induced inhibition (Supplementary Text 3). This lack of correlation supports their exclusion from our model.

(5) Limitations: We acknowledge the inherent **limitations** and **uncertainties** associated with our model development and subsequent extrapolations: 1) photodegradation impact: our experiment excluded photodegradation, which is a key factor in regulating MeHg dynamics in oligotrophic or clear-water lakes. This omission may affect the accuracy of absolute MeHg level predictions for these lake types; 2) microbial acclimation: we quantified the responses of microbial Hg methylation to elevated CO₂ levels over a time span of two months. Further investigation is needed to quantify the long-term responses, as microbes including methanogens, could gradually acclimated to environmental changes with prolonged incubation time; 3) Joint effects of multiple climatic factors: We focused solely on the effects of elevated

CO₂ concentrations and rising temperature on MeHg production. However, climate change encompasses a range of factors, including precipitation, ultraviolet radiation, and other variables, which may interact in complex ways. The influence of these additional climatic factors, as well as potential multi-factor coupling, could lead to significant variations in Hg cycling pathways within lakes; 4) DOC Heterogeneity: In our experiment, the inhibitory effect of elevated CO₂ on MeHg production was more pronounced as the proportion of AOM in the DOC pool increased. This suggests that our model may overestimate MeHg reductions in lakes with high DOC concentrations dominated by terrestrial DOM.

To address your concerns and to ensure scientific rigor, we have revised the Methods to include Supplementary Table 4 (raw regression data) and clarified the potential limitations, emphasizing the model's robustness for eutrophic systems where DOC/MeHg levels align with our experimental range (Page S5-S6, Line 89-106).

“Our model incorporates DOC as a parameter primarily due to the availability of global DOC datasets, which enable broad-scale extrapolation. However, DOC represents a heterogeneous pool encompassing both autochthonous and allochthonous fractions that may differentially regulate Hg methylation and demethylation⁹. Specifically, algae-derived AOM, particularly in eutrophic lakes, is a functionally distinct driver of microbial Hg methylation¹⁰, while soil-derived SOM typically exerts weaker effects⁹. To clarify how AOM contributions modulate CO₂-driven inhibition of MeHg production, we conducted additional experiments with mixed DOM sources: pure AOM, pure SOM, and AOM:SOM ratios of 80:20, 50:50, and 20:80. Elevated CO₂ significantly suppressed net MeHg production only when AOM constituted \geq 50% of the DOM pool ($p < 0.001$). The inhibition rate declined from 75% (pure AOM) to 68% (80:20), 52% (50:50), and became negligible (<10%) when AOM was <50% (Supplementary Fig. 4). These findings demonstrate that while our model may overestimate MeHg declines in lakes dominated by terrestrial DOM, the deviation for eutrophic lakes (the principal focus of this study) is moderate. After normalizing by the AOM:SOM ratio, projected global inhibition ranges were refined from 31–86% (mean 50%) to 33–74% (mean 47%).”

Supplementary Table 4 Data used in the multiple linear regression analysis

No. ^a	DOC/mg·L ⁻¹	Dissolved MeHg/ng·L ⁻¹	Reduction rate	N-Reduction rate ^b
1	2.055	0.939	0.402	0.402
2	4.716	0.862	0.474	0.474
3	2.774	0.834	0.434	0.434
4	4.115	1.073	0.363	0.363
5	3.649	0.889	0.323	0.323

6	6.301	0.902	0.267	0.267
7	3.395	0.922	0.386	0.386
8	4.600	0.849	0.402	0.402
9	3.483	0.784	0.398	0.398
10	4.039	0.813	0.389	0.389
11	7.788	0.846	0.487	0.487
12	5.549	0.852	0.379	0.379
13	8.149	0.836	0.437	0.437
14	8.254	0.856	0.496	0.496
15	5.886	0.808	0.514	0.514
16	5.753	1.019	0.572	0.572
17	4.741	0.479	0.528	0.528
18	4.462	0.347	0.485	0.485
19	4.522	0.452	0.452	0.452
20	6.297	1.070	0.525	0.525
21	5.102	0.947	0.504	0.504
22	4.448	0.934	0.448	0.448
23	8.975	0.756	0.441	0.441
24	2.799	0.703	0.554	0.554
25	5.892	0.611	0.550	0.550
26	5.215	0.643	0.526	0.526
27	4.371	0.538	0.578	0.578
28	4.578	0.920	0.386	0.386
29	8.897	1.099	0.295	0.295
30	6.406	0.988	0.415	0.415
31	4.949	1.149	0.465	0.465
32	8.852	1.085	0.395	0.395
33	7.587	0.744	0.284	0.284
34	5.546	0.894	0.286	0.286
35	8.337	0.854	0.168	0.168
36	9.420	0.788	0.253	0.253
37	6.182	0.616	0.603	0.603
38	5.248	0.328	0.282	0.282
39	7.483	0.229	0.433	0.433
40	8.611	0.225	0.247	0.247
41	6.010	0.295	0.225	0.225
42	6.918	0.471	0.267	0.267
43	7.711	0.805	0.238	0.238
44	4.450	0.738	0.419	0.419
45	6.368	0.796	0.141	0.141
46	4.934	0.888	0.356	0.356
47	5.233	0.894	0.416	0.416
48	3.843	0.943	0.318	0.318
49	9.254	35.865	0.936	0.838

50	12.949	39.297	0.892	0.798
51	10.253	30.367	0.935	0.837
52	12.115	13.161	0.944	0.845
53	11.468	21.810	0.880	0.788
54	15.151	10.903	0.730	0.653
55	11.115	12.714	0.843	0.755
56	12.788	39.011	0.948	0.848
57	11.238	18.636	0.931	0.833
58	12.009	23.330	0.927	0.830
59	17.216	18.891	0.917	0.821
60	14.107	22.180	0.934	0.836
61	17.718	24.576	0.905	0.810
62	17.864	23.027	0.914	0.817
63	14.575	2.877	0.544	0.486
64	14.390	1.904	0.656	0.587
65	12.985	7.425	0.851	0.761
66	12.597	2.424	0.798	0.714
67	12.681	11.245	0.915	0.819
68	15.146	23.711	0.912	0.816
69	13.486	18.137	0.873	0.782
70	12.578	15.161	0.827	0.740
71	18.865	18.462	0.819	0.733
72	10.288	26.538	0.814	0.729
73	14.583	13.358	0.817	0.731
74	13.643	9.541	0.699	0.625
75	12.471	21.715	0.919	0.822
76	12.758	18.811	0.923	0.826
77	18.757	14.620	0.829	0.741
78	15.297	21.647	0.913	0.817
79	13.274	15.906	0.894	0.800
80	18.694	16.423	0.858	0.768
81	16.937	24.433	0.958	0.858
82	14.102	24.044	0.915	0.819
83	17.978	24.323	0.899	0.804
84	19.483	21.409	0.847	0.758
85	14.986	18.101	0.868	0.777
86	13.689	5.844	0.865	0.774
87	16.792	6.841	0.820	0.733
88	18.359	7.910	0.838	0.750
89	14.747	16.998	0.789	0.706
90	16.008	8.317	0.687	0.614
91	17.109	16.862	0.850	0.761
92	12.580	26.090	0.694	0.621
93	15.244	28.929	0.932	0.834

94	13.253	19.332	0.876	0.784
95	13.668	24.322	0.887	0.794
96	11.737	7.472	0.833	0.746
97	13.844	4.493	0.888	0.888
98	25.354	28.929	0.932	0.834

^a 1-48 and 49-96 represented the results for the lake water control group and “+Algae” treatment group along the middle and lower reaches of the Yangtze River. 97-98 represented the results for Chaohu Lake water and its “+ algae” treatment group from the mechanism investigation experiment. The DOC in “+ Algae” treatments can be estimated as: $[\text{DOC}]_{\text{Lake water} + \text{algal biomass}} = (1.389 \pm 0.075) \times [\text{DOC}]_{\text{lake water}} + (6.4 \pm 0.638)$ ($R^2_{\text{adjust}} = 0.991$, $p = 0.003$, see Supplementary, Fig. 12).

^b 49-96 and 98 represented the values of the inhibition rates of MeHg net production in the “+Algae” treatments used in the model after normalization according to the “M1/AOM” ratio (i.e., 89.5%, where M1 represents the mixture of 20% terrestrial organic matter and 80% AOM)

Novelty of your research

Your findings that increasing CO₂ shifted the methanogen community from acetoclastic (more significant methylators) to hydrogenotrophic (less methylation capabilities) was very compelling and helpful for the scientific community. I think you should comment on the energetics and biochemistry of hydrogenotrophic vs acetoclastic metabolism more. Hydrogenotrophic metabolism produces more energy per mole of methane produced and is preferred if all substrates are equal. However, the law of the limiting factor suggests that it is much more complex than simply carbon dioxide concentration – which should not often be limited.

Note that 4 hydrogen molecules are needed per molecule of carbon dioxide. In anaerobic respiration, approximately 2 moles of H₂ are produced per mole of acetate. So as AOM is broken down, hydrogen will be limited compared to acetate and carbon dioxide and hydrogen production would need to be increased in order to stimulate hydrogenotrophic metabolism. I suspect fermenting organisms that produce more hydrogen than acetate were involved, since the incubations were performed in the dark and because you aerated the control (which introduces oxygen), but only added CO₂ to the treatments – they may not have had enough oxygen for redox compound cycling (Fe³⁺ -> Fe²⁺ -> Fe³⁺; SO₄ -> H₂S -> SO₄, etc). There are also algae, cyanobacteria, and green/purple sulfur bacteria that produce hydrogen during photosynthesis. Since they would be stimulated by carbon dioxide (especially in 2 month incubation), they could increase the hydrogen concentration needed to

encourage hydrogenotrophic growth and although your experiment didn't look at this, it is a reasonable connection to your results - increasing CO₂ would encourage them and increase hydrogen and carbon dioxide, ultimately selecting for hydrogenotrophic metabolism.

Response: We sincerely appreciate your theoretical analysis of CO₂-mediated shifts in methanogen communities, particularly your insights from energy and hydrogen perspectives. These insights have significantly advanced our mechanistic understanding of these processes at molecular scales. Particularly, your conceptual framework addressing substrate limitations and metabolic trade-offs has provided a critical foundation for interpreting our experimental findings.

The core distinction between hydrogenotrophic and acetoclastic methanogenesis lies in their energy yields and substrate requirements. Hydrogenotrophic methanogens generate methane via $\text{CO}_2 + 4 \text{H}_2 \rightarrow \text{CH}_4 + 2 \text{H}_2\text{O}$, a pathway that produces more ATP per mole of methane (approximately 1.5–2.5 ATP) compared to acetoclastic metabolism $\text{CH}_3\text{COOH} \rightarrow \text{CH}_4 + \text{CO}_2$, which yields only ~1 ATP per mole^{24, 25}. This energetic advantage explains why hydrogenotrophy is favored when substrates (especially H₂ and CO₂) are abundant. However, as noted, H₂ availability is often limiting: acetate fermentation typically produces only 2 moles of H₂ per mole of acetate, creating a bottleneck for hydrogenotrophic growth in natural systems.

In our experiments, the observed shift toward hydrogenotrophic methanogens (e.g., *Methanobacterium*) under elevated CO₂ likely reflects a reorganization of substrate fluxes. Algal organic matter (AOM) degradation by fermentative bacteria—such as *Clostridia* and *Enterobacter*—would have generated both acetate and H₂ via dark fermentation (e.g., glucose → 2 acetate + 2 CO₂ + 4 H₂). Elevated CO₂ may have stimulated these fermenters to prioritize H₂ production over acetate, **alleviating H₂ limitation** and enabling hydrogenotrophs to outcompete acetoclastic populations (e.g., *Methanosarcina*) despite their lower energy yield per substrate. This aligns with your hypothesis that fermenting organisms play a pivotal role in mediating this shift in methanogen communities.

We acknowledge your emphasis on H₂ as a key regulator and regret that our attempts to quantify headspace H₂ were unsuccessful due to technical constraints: the combination of hydrogen's low solubility and rapid microbial consumption by methanogens created substantial methodological challenges for accurate H₂ quantification in our microcosm experiments. However, the consistent shift toward hydrogenotrophs across all treatments may indirectly suggest that H₂ availability remained sufficient to sustain methanogenic activity

under elevated CO₂ conditions.

Regarding photosynthetic H₂ production: our **2-month** dark incubation explicitly excluded this pathway, as algae and cyanobacteria cannot perform photosynthesis in the absence of light. Thus, H₂ in our system must have originated from dark fermentation of AOM, reinforcing the role of fermentative bacteria as the primary H₂ source.

To address your concerns, we have revised the discussion to incorporate these perspectives in Supplementary Text 2, emphasizing the interplay between fermentative H₂ production and methanogenic pathways in driving the observed shifts (Page S7-S8, Line 124-156).

“In the process of methanogenesis, H₂ plays a crucial role, and its availability is a key factor regulating the community structure of hydrogenotrophic and acetoclastic methanogens²⁶. Hydrogenotrophic methanogenesis occurs through the reaction $\text{CO}_2 + 4 \text{H}_2 \rightarrow \text{CH}_4 + 2 \text{H}_2\text{O}$, producing approximately 1.5–2.5 ATP per mole of methane. In contrast, acetoclastic methanogenesis ($\text{CH}_3\text{COOH} \rightarrow \text{CH}_4 + \text{CO}_2$) yields only about 1 ATP per mole^{24, 25}. Therefore, hydrogenotrophic methanogenesis exhibits advantages in energy output and is more favored when substrates such as H₂ and CO₂ are abundant. However, the supply of H₂ is often limited because acetate fermentation typically produces only 2 moles of H₂ per mole, which becomes a bottleneck for the growth of hydrogenotrophic methanogens in natural systems²⁷.

In this study, the observed shift of the methanogen community towards hydrogenotrophic types (e.g., *Methanobacterium*) under high CO₂ conditions may be since high CO₂ stimulates fermentative bacteria such as *Clostridia* and *Enterobacter* to degrade algal organic matter. Through dark fermentation (e.g., $\text{glucose} \rightarrow 2 \text{ acetate} + 2 \text{ CO}_2 + 4 \text{ H}_2$), the substrate flux is adjusted to favor H₂ production over acetate, thereby alleviating H₂ limitation and enabling hydrogenotrophic methanogens to outcompete acetoclastic populations (e.g., *Methanosarcina*). Nevertheless, the combination of hydrogen’s low solubility and rapid microbial consumption by methanogens created substantial methodological challenges for accurate H₂ quantification in our microcosm experiments. The consistent shift towards hydrogenotrophic methanogens across all treatments suggests that H₂ availability was sufficient to support their growth under high CO₂ conditions. Given the critical role of H₂, future research should focus on the following aspects: using more sensitive detection methods to track H₂ dynamics in real-time to quantify its coupling with changes in the methanogen community; identifying key fermentative taxa responsible for H₂ production under high CO₂ conditions; and measuring ATP yields in pure

cultures of hydrogenotrophic and acetoclastic methanogens to directly verify energy trade-offs, thereby further clarifying the role and mechanism of H₂ in methanogenesis and community structure regulation under climate change.”

Minor Comments

Figure 2 – amazing graph, but it’s difficult to interpret that it is ~90% reduction due to the way it is shown and the label. I would suggest having the Y axis be 0% at the top and -100% at the bottom, with the larger bars further downwards. This emphasized that CO₂ had a negative impact on methylation. The Y axis label is kind of confusing, I would suggest “MeHg production compared to the control” and then the negative numbers will highlight that increased CO₂ led to decreased MeHg production.

Response: We appreciate your constructive feedback on improving the clarity of Figure 2, which aims to communicate the inhibitory effect of elevated CO₂ more effectively on MeHg production. To maintain consistency with the manuscript’s use of **“Inhibited MeHg production by elevated CO₂”** terminology, we presented the Y-axis scale as 0%-100% (rather than 0%-[100%]) throughout all figures. To ensure clarity, we have added explanatory information to clarify the axis labeling as follows: “Inhibited MeHg production by elevated CO₂ are denoted as the percentage changes of MeHg levels between “Ambient CO₂” and “Elevated CO₂”. For enhanced clarity, we present both inhibition rates (Fig. 2) and MeHg concentrations (Supplementary Fig. 2).

Fig. 2 Elevated CO₂ inhibited MeHg production under different scenarios. (a) Inhibition rate of MeHg production under different elevated CO₂ levels (650 or 1000 ppm) compared to ambient 420 ppm CO₂. (b) Inhibition rate of MeHg production by elevated CO₂ (1000 ppm) compared to ambient CO₂ (420 ppm) under bloom intensities (200 to 2000 $\mu\text{g}\cdot\text{L}^{-1}$ chlorophyll a). (c) Inhibition rate of MeHg production by elevated CO₂ (1000 ppm) compared to ambient CO₂ (420 ppm) under different stages of algal decomposition. “Lake water”: unfiltered lake water. “+200 Algae”, “+1000 Algae”, and “+2000 Algae”: unfiltered lake water (40 mL) amended with 0.001, 0.005, and 0.01 g of algal biomass, respectively. Before sampling on “Day 0”, all lake water was equilibrated for 4 h with Hg(II). Inhibited MeHg production by elevated CO₂ are denoted as the percentage changes of MeHg levels between “Ambient CO₂” and “Elevated CO₂”. Error bars show standard errors of the mean ($n=3$ or 4).

Supplementary Fig. 2 Net MeHg production in different scenarios under ambient (420 ppm) or elevated CO₂ (1000 ppm) conditions. (a) Different CO₂ levels; (b) Different trophic statuses; (c) Different decomposition stages. “Lake water”: unfiltered lake water. “+200 Algae”, “+1000 Algae”, and “+2000 Algae”: unfiltered lake water (40 mL) amended with 0.001, 0.005, and 0.01 g of algal biomass, respectively. Before sampling on “Day 0”, all lake water was equilibrated for 4 h with Hg(II). Different lowercase letters above the bars indicate significant differences among different treatments ($p < 0.05$). The asterisk above the bars indicates a significant difference between “420 ppm CO₂” and “1000 ppm CO₂” (*: $p < 0.05$; **: $p < 0.01$; ***: $p < 0.001$; without *: $p > 0.05$).

Line 340 – change to “collected using a plankton net and stored at 4 °C”

Response: Revised as suggested.

Line 347 – change experiment to experiments

Response: Revised as suggested.

Line 351 – add “of” inbetween “the mechanisms of elevated CO₂-impacted”

Response: Revised as suggested.

Line 363 – is this supposed to be 20 ml?

Response: A 2 ml sample volume was used because post-7-day-incubation MeHg concentrations exceeded 0.2 ng·L⁻¹ across all treatment groups. Given the calibration curve range of 0.5-50 pg, this loading volume ensures accurate quantification. To ensure clarity, we have added an explanation (Page 23, Line 442-444): “A 2 mL sample volume was selected because, after the 7-day incubation, MeHg concentrations exceeded 0.2 ng·L⁻¹ in all treatments. Within the calibration curve range of 0.5–50 pg, this volume ensured accurate quantification.”

Line 386 – “the” is typed twice before observed, remove one.

Response: Thanks for identifying this typo. Revised as suggested.

Supplemental Material

Fig T3-2 – adjust the X-axis so that it starts at 0.0. It is slightly off and starts < 0.0.

Response: Revised as suggested.

Supplementary Fig. T4-2 The levels of dissolved MeHg in the water from global lakes.

References

1. Zhang, Y.; Dutkiewicz, S.; Sunderland, E. M., Impacts of climate change on methylmercury formation and bioaccumulation in the 21st century ocean. *One Earth* **2021**, *4*, (2), 279-288.
2. IPCC *Climate Change 2023: Sixth Assessment Report (AR6). Intergovernmental Panel on Climate Change*; 2023; p <https://www.ipcc.ch/report/ar6/syrl/>.
3. Yang, Z.; Fang, W.; Lu, X.; Sheng, G.-P.; Graham, D. E.; Liang, L.; Wullschleger, S. D.; Gu, B., Warming increases methylmercury production in an Arctic soil. *Environmental Pollution* **2016**, *214*, 504-509.
4. Barkay, T.; Gu, B., Demethylation - the other side of the mercury methylation coin: A critical review. *ACS Environmental Au* **2021**, *2*, (2), 77–97.
5. Lei, P.; Zhu, J.; Zhang, J.; He, H.; Chen, M.; Zhong, H., Algal organic matter inhibits methylmercury photodegradation in eutrophic lake water: A dynamic study. *Science of the Total Environment* **2023**, *899*, 165661.
6. Eriksson, T.; Öquist, M. G.; Nilsson, M. B., Production and oxidation of methane in a boreal mire after a decade of increased temperature and nitrogen and sulfur deposition. *Global Change Biology* **2010**, *16*, (7), 2130-2144.
7. Holm, S.; Walz, J.; Horn, F.; Yang, S. Z.; Grigoriev, M. N.; Wagner, D.; Knoblauch, C.; Liebner, S., Methanogenic response to long-term permafrost thaw is determined by paleoenvironment. *FEMS Microbiology Ecology* **2020**, *96*, (3), fiae021.
8. Liu, S. S.; He, Z. Q.; Tang, Z.; Liu, L. Z.; Hou, J. W.; Li, T. T.; Zhang, Y. H.; Shi, Q.; Giesy, J. P.; Wu, F. C., Linking the molecular composition of autochthonous dissolved organic matter to source identification for freshwater lake ecosystems by combination of optical spectroscopy and FT-ICR-MS analysis. *Science of the Total Environment* **2020**, *703*, 134764.
9. Wu, Z. Y.; Li, Z. K.; Shao, B.; Chen, J.; Cui, X. M.; Cui, X. Y.; Liu, X. H.; Zhao, Y. X.; Pu, Q.; Liu, J.; He, W.; Liu, Y. W.; Liu, Y. R.; Wang, X. J.; Meng, B.; Tong, Y. D., Differential response of Hg-methylating and MeHg-demethylating microbiomes to dissolved organic matter components in eutrophic lake water. *Journal of Hazardous Materials* **2024**, *465*, 133298.
10. Lei, P.; Zhang, J.; Zhu, J.; Tan, Q.; Raymond, W. M. K.; Pan, K.; Jiang, T.; Mohammad, N.; Zhong, H., Algal organic matter drives methanogen-mediated methylmercury production in water from eutrophic shallow lakes. *Environmental Science & Technology* **2021**, *55*, (15), 10811-10820.
11. Sun, R. Y.; Hintelmann, H.; Wiklund, J. A.; Evans, M. S.; Muir, D.; Kirk, J. L., Mercury isotope variations in lake sediment cores in response to direct mercury emissions from non-ferrous metal smelters and legacy mercury remobilization. *Environmental Science & Technology* **2022**, *56*, (12), 8266-8277.
12. Xujun Liang; Huan Zhong; Alexander Johs; Pei Lei; Jin Zhang; Neslihan Taş; Lijie

- Zhang; Linduo Zhao; Nali Zhu; Xixiang Yin; Lihong Wang; Eddy Y. Zeng; Yuxi Gao; Jiating Zhao; Dale A. Pelletier; Eric M. Pierce; Gu, B., Light-independent phytoplankton degradation and detoxification of methylmercury in water. *Nature Water* **2023**, *1*, 705-715.
13. Chetelat, J.; Cloutier, L.; Amyot, M., An investigation of enhanced mercury bioaccumulation in fish from offshore feeding. *Ecotoxicology* **2013**, *22*, (6), 1020-1032.
 14. Li, Y. B.; Cai, Y., Progress in the study of mercury methylation and demethylation in aquatic environments. *Chinese Science Bulletin* **2013**, *58*, (2), 177-185.
 15. Taylor, V. F.; Buckman, K. L.; Seelen, E. A.; Mazrui, N. M.; Balcom, P. H.; Mason, R. P.; Chen, C. Y., Organic carbon content drives methylmercury levels in the water column and in estuarine food webs across latitudes in the Northeast United States. *Environmental Pollution* **2019**, *246*, 639-649.
 16. Lei, P.; Zhou, S.; Kong, Y.; Zhang, J.; He, H.; Zhong, H., Response of mercury methylation to algal bloom decomposition or elevated CO₂ in surface sediments from the East China Sea. *Environmental Pollution* **2025**, *383*, 126786.
 17. Lyon, B. F.; Ambrose, R.; Rice, G.; Maxwell, C. J., Calculation of soil-water and benthic sediment partition coefficients for mercury. *Chemosphere* **1997**, *35*, (4), 791-808.
 18. Lei, P.; Nunes, L. M.; Liu, Y.-R.; Zhong, H.; Pan, K., Mechanisms of algal biomass input enhanced microbial Hg methylation in lake sediments. *Environment International* **2019**, *126*, 279-288.
 19. Li, Y.; Cai, Y., Progress in the study of mercury methylation and demethylation in aquatic environments. *Chinese Science Bulletin* **2013**, *58*, (2), 177-185.
 20. Shi, X.; Zhao, X.; Zhang, M.; Yang, Z.; Xu, P.; Kong, F., The responses of phytoplankton communities to elevated CO₂ show seasonal variations in the highly eutrophic Lake Taihu. *Canadian Journal of Fisheries and Aquatic Sciences* **2015**, *73*, (5), 727-736.
 21. Beckers, F.; Rinklebe, J., Cycling of mercury in the environment: Sources, fate, and human health implications: A review. *Critical Reviews in Environmental Science and Technology* **2017**, *47*, (9), 693-794.
 22. Qiu, G.; Feng, X.; Wang, S.; Shang, L., Environmental contamination of mercury from Hg-mining areas in Wuchuan, northeastern Guizhou, China. *Environmental Pollution* **2006**, *142*, (3), 549-558.
 23. Wu, Z.; Li, Z.; Shao, B.; Zhang, Y.; He, W.; Lu, Y.; Gusvitskii, K.; Zhao, Y.; Liu, Y.; Wang, X.; Tong, Y., Impact of dissolved organic matter and environmental factors on methylmercury concentrations across aquatic ecosystems inferred from a global dataset. *Chemosphere* **2022**, *294*, 133713.
 24. Conrad, R., Importance of hydrogenotrophic, acetoclastic and methylotrophic methanogenesis for methane production in terrestrial, aquatic and other anoxic environments: A mini review. *Pedosphere* **2020**, *30*, (1), 25-39.

25. Demirel, B.; Scherer, P., The roles of acetotrophic and hydrogenotrophic methanogens during anaerobic conversion of biomass to methane: a review. *Reviews in Environmental Science and Bio/Technology* **2008**, *7*, (2), 173-190.
26. Kakuk, B.; Wirth, R.; Maróti, G.; Szuhaj, M.; Rakhely, G.; Laczi, K.; Kovács, K. L.; Bagi, Z., Early response of methanogenic archaea to H₂ as evaluated by metagenomics and metatranscriptomics. *Microbial Cell Factories* **2021**, *20*, (1), PMC8254922.
27. Szuhaj, M.; Acs, N.; Tengölics, R.; Bodor, A.; Rákhely, G.; Kovács, K. L.; Bagi, Z., Conversion of H₂ and CO₂ to CH₄ and acetate in fed-batch biogas reactors by mixed biogas community: a novel route for the power-to-gas concept. *Biotechnology for Biofuels* **2016**, *9*, 102.

Response to reviewers

For clarity, we present reviewer comments in *italics*, our response to comments in regular text, and a description of the changes to the manuscript in underlined text. Revisions in the revised manuscript and supplementary information are highlighted in **yellow**.

Reviewer #1:

This paper considers the question of how a warmer climate with higher atmospheric CO₂ concentrations and algal blooms will affect the concentrations and net production of neurotoxic methylmercury in lakes. This is a relevant question, and the paper presents an ambitious, well-executed set of experiments to answer that question. The authors have responded very well to an earlier round of reviews with new experiments and revisions to address the first set of reviews. I think this manuscript could be a valuable addition to the literature that is appropriate to the readership of Nature Communications.

Response: We sincerely appreciate your positive evaluation and constructive comments. Your feedback has helped us further improve the clarity, balance, and scientific accuracy of the manuscript.

I have just two minor concerns that I suggest the authors' address.

The first concern is that the paper claims that the findings show that elevated CO₂ will counterbalance the effect of algal blooms on MeHg concentrations in lake, and will help stabilize MeHg levels. I think this suggestion of stability in future MeHg levels is claiming somewhat more than is possible given the complexity of mercury methylations. Claiming a little less would be more convincing in my ears. For instance, in the abstract (line 57), to write "counteract" rather than "counterbalance" would be more appropriate. I also don't think it is appropriate to speak of 'helping to stabilize future MeHg concentrations' (paraphrasing lines 59-60). To speak about counteracting is quite important enough.

Response: We sincerely thank you for this thoughtful suggestion. We fully agree that, given the complex and nonlinear nature of Hg methylation and demethylation processes, future trends in MeHg levels should be described with more caution. Following your recommendation, we have revised the abstract and all relevant sections to replace "counterbalance" (Abstract line 53) and change the statement of "stabilize" by "reduce uncertainties in predicting MeHg risks." (Results and discussion line 108-109; 399-400). This wording more accurately reflects our findings: elevated CO₂ substantially attenuated the MeHg increase induced by algal organic matter, but does not imply long-term stabilization of MeHg concentrations. We believe this revised phrasing avoids overinterpretation while preserving the key scientific message.

The other minor concern is that I tried to look for what the effect of adding 1000 ug/L of algal organic matter without changing the CO₂ levels was in the experiments. That information may be there, but I think this effect deserves to be more clearly stated in the text (apologies if I missed that). I could locate in diagrams the experimental effect of increasing CO₂ without increasing algal AOM, but those results would also be worth making clear, as a complement to the focus on the effect of combining increases in algal organic matter and CO₂.

Response: Thank you for raising this important point. We agree that the effect of algal organic matter (AOM) addition under ambient CO₂ conditions should be presented more explicitly in the main text to complement the combined treatment results. In our experiments, AOM addition alone led to a substantial increase in MeHg concentrations, with increases ranging from 86% to 4592% across 45 eutrophic lakes in the middle and lower Yangtze River basin, yielding an average increase of 2345% (Supplementary Fig. 1a&c). This strong enhancement is consistent with our previous findings, e.g., Lei, P. et al., *Environmental Science & Technology*, 2021, 55(15), 10811-10820, which demonstrated that algal-derived organic matter stimulates methanogen abundance and activity, thereby promoting MeHg production in eutrophic lake waters. In the original submission, we summarized this outcome briefly to maintain focus on the combined effects of climate drivers. Following your suggestion, we have explicitly stated the effect of AOM addition alone in the Results and discussion (lines 121–124): “Under ambient CO₂ levels, AOM markedly enhanced MeHg production, yielding 1–3 orders of magnitude higher levels across 45 lakes (Supplementary Fig. 1), likely driven by stimulated methanogenesis-mediated MeHg production”. These revisions make the independent role of AOM clearer and strengthen the logical progression of our argument regarding interactive climate effects.

Supplementary Fig. 1 Elevated CO₂ (1000 ppm) decreased the levels of MeHg in water from lakes along the middle and lower reaches of the Yangtze River. (a) the levels of MeHg in “Lake water” groups under 420 ppm CO₂. (b) the levels of MeHg in “Lake water” groups under 1000 ppm CO₂. (c) the levels of MeHg in “+ 1000 Algae” treatments under 420 ppm CO₂. (d) the levels of MeHg in “+ 1000 Algae” treatments under 1000 ppm CO₂. The sampling time was set to Day 7. “+ 1000 Algae”: unfiltered lake water (40 mL) amended with 0.005 g of algal biomass from Chaohu Lake. “Lake water”: unfiltered lake water with no algal addition. The map of the middle and lower reaches of the Yangtze River was generated using 1:1,000,000 National Fundamental Geographic Information Data of China (<https://www.webmap.cn/commres.do?method=result100W>).